# How I Warped Your Noise: a Temporally-Correlated Noise Prior for Diffusion Models

**Pascal Chang, Jingwei Tang, Markus Gross, Vinicius C. Azevedo**

`<firstname>.<lastname>@inf.ethz.ch`

ETH Zürich, Switzerland

## Abstract

Video editing and generation methods often rely on pre-trained image-based diffusion models. During the diffusion process, however, the reliance on rudimentary noise sampling techniques that do not preserve correlations present in subsequent frames of a video is detrimental to the quality of the results. This either produces high-frequency flickering, or texture-sticking artifacts that are not amenable to post-processing. With this in mind, we propose a novel method for preserving temporal correlations in a sequence of noise samples. This approach is materialized by a novel noise representation, dubbed $\int$-noise (integral noise), that reinterprets individual noise samples as a continuously integrated noise field: pixel values do not represent discrete values, but are rather the integral of an underlying infinite-resolution noise over the pixel area. Additionally, we propose a carefully tailored transport method that uses $\int$-noise to accurately advect noise samples over a sequence of frames, maximizing the correlation between different frames while also preserving the noise properties. Our results demonstrate that the proposed $\int$-noise can be used for a variety of tasks, such as video restoration, surrogate rendering, and conditional video generation. See `https://warpyournoise.github.io` for video results.

## 1 Introduction

Despite their notable ability to generate high-quality images (Rombach et al., 2022; Nichol et al., 2022) from simple text inputs, diffusion models (Sohl-Dickstein et al., 2015; Ho et al., 2020) have been applied with limited success when it comes to video processing and generation. While it is a common practice to build diffusion-based video pipelines upon pretrained text-to-image models (Ceylan et al., 2023; Geyer et al., 2023; Yang et al., 2023; Khachatryan et al., 2023), these approaches do not have a systematic way to preserve the natural correlations between subsequent frames from a video. One contributing factor to this problem is the indiscriminate use of noise to corrupt data during inference: all relevant motion information present in a video is lost in the diffusion process. Employing a different set of noises for each frame within video sequences yields results that manifest high-frequency hallucinations that are challenging to eliminate, and some approaches resort to fine-tuning neural networks with temporal attention layers (Liew et al., 2023; Liu et al., 2023b; Shin et al., 2023; Zhao et al., 2023; Wu et al., 2022).

Other than using random sets of noise at inference, it is also common to employ fixed noise as a content-agnostic solution for artificially enforcing correlations. However, the problem of fixed noise signals manifests as texture-sticking artifacts that are oblivious to post-process techniques. Alternatively, one can also invert the set of noises that reconstruct a given image sequence under a conditional prompt (Ceylan et al., 2023; Geyer et al., 2023). The limitation of such an approach is that the set of inverted noises closely entangles the temporal information with the content of the frames, making such methods not suitable for injecting the learned temporal correlation into arbitrary noise samples. Lastly, techniques such as feature warping (Ni et al., 2023; Yang et al., 2023), or cross-frame attention (Ceylan et al., 2023; Geyer et al., 2023; Yang et al., 2023; Khachatryan et al., 2023) can also alleviate temporal coherency issues to a certain extent. However, these approaches are limited because the features are not able to represent high frequencies patterns of the fine resolution image. Given the importance of the role of the noise in inferring diffusion models,

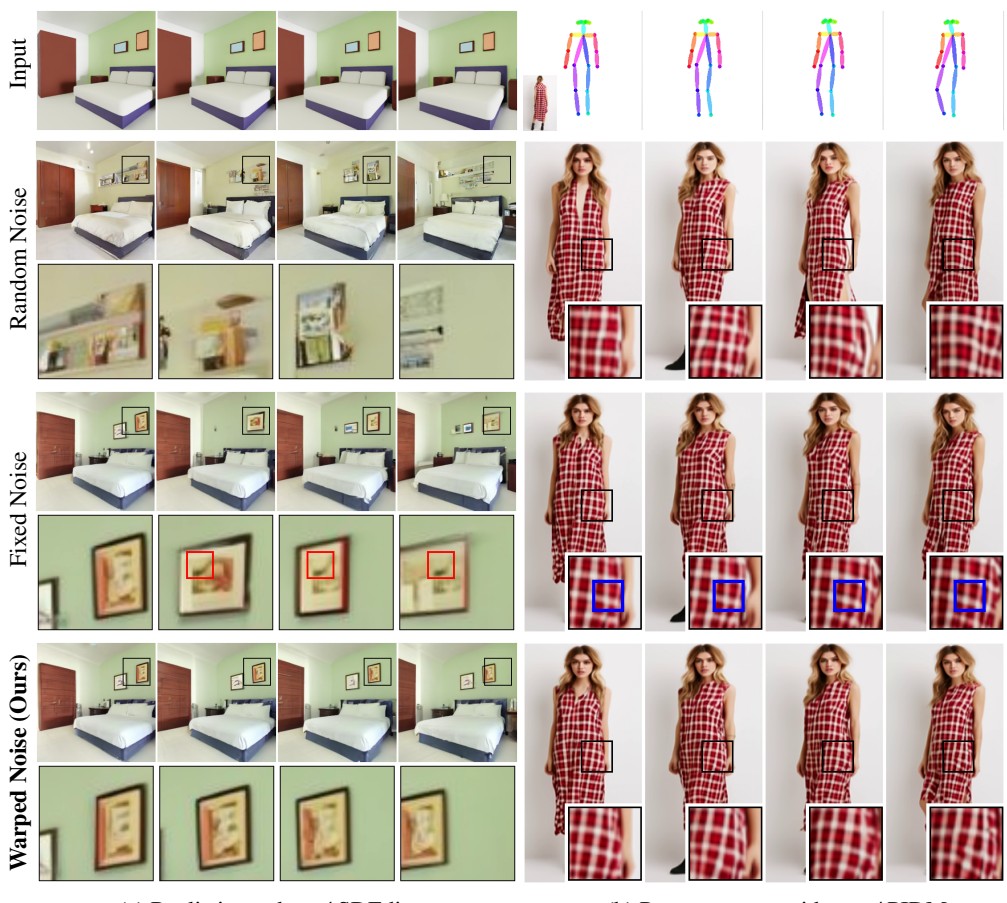

(a) Realistic render w/ SDEdit          (b) Pose-to-person videos w/ PIDM

Figure 1: Our noise warping method lifts diffusion-based image editing methods like SDEdit (Meng et al., 2022) and Person Image Diffusion Model (PIDM) (Bhunia et al., 2022) to the temporal domain. It avoids unnatural flickering and texture sticking artifacts (see colored squares) that commonly appears with standard noise priors.

it's surprising how little research was devoted to exploring the impact of noise priors on temporal coherency. Thus, we were motivated by the simple question: how can we create a noise prior that preserves correlations present in an image sequence?

Our answer is to warp a noise signal, such that it generates a new noise sample that preserves the correlations induced by motion vectors or optical flow. However, implementing such approach is not so simple. The commonly employed Gaussian noise has special properties that are lost when warped/transported with standard methods. Transport relies on the sub-sampling of a noise signal in an undeformed space. These sampling operations create important cross-correlations that are detrimental to the preservation of the noise power spectral density and to the independence between pixels within a single sample. Our paper addresses the outlined limitations in transporting noise fields with three novel contributions. The first one is to reinterpret individual noise samples used in diffusion models as a continuously integrated noise field: pixel values do not represent discrete values, but are rather the integral of an underlying infinite noise. We dub this new interpretation of a discrete noise sample as $\int$-noise (integral noise). The second contribution is the derivation of the *noise transport equation*, which allows accurate, distribution-preserving transport of a continuously defined noise signal. Lastly, we design a carefully tailored transport algorithm that discretizes the noise transport equation, maximizing the temporal correlation between samples while also preserving its original properties. Our results demonstrate that our noise warping method creates temporally-correlated noise fields, which can be used for a variety of tasks such as video restoration and editing, surrogate rendering, and conditional video generation.

## 2 METHOD

### 2.1 THE $\int$-NOISE REPRESENTATION

A discrete 2D Gaussian noise of dimension $D \times D$ is represented by the function $G : (i,j) \in \{1, \ldots, D\}^2 \to X_{i,j}$ that maps a pixel coordinate $(i,j)$ to a random variable $X_{i,j}$. Diffusion models employ this discrete formulation, and random variables are assumed to be independently and identically distributed (i.i.d.) Gaussian samples $X_{i,j} \sim \mathcal{N}(0,1)$. At the core of our work is the reinterpretation of this discrete 2D Gaussian noise as the integral of an underlying infinite noise field.

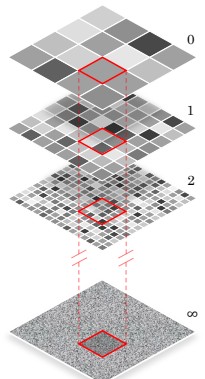

An infinite-resolution noise field is represented by a 2D white Gaussian noise signal. To construct that, we start by endowing the domain $E = [0,D] \times [0,D]$ with the usual Borel $\sigma$-algebra $\mathcal{E} = \mathcal{B}(E)$ and the standard Lebesgue measure $\nu$. Then, the white Gaussian noise on the $\sigma$-finite measure space $(E, \mathcal{E}, \nu)$ is defined as a function $W : A \in \mathcal{E} \to W(A) \sim \mathcal{N}(0, \nu(A))$ that maps $A$ — a subset of the domain $E$ — to a Gaussian-distributed variable, with variance $\nu(A)$ (Walsh, 2006). There are many valid ways to subdivide the domain representing the continuous noise. The standard discrete setting partitions the domain $E$ into $D \times D$ regularly spaced non-overlapping square subsets. We denote this partition as $\mathbb{A}^0 \subseteq \mathcal{E}$ (level 0 in the inset image). Another way of partitioning the domain is to further refine $E$ into a higher resolution set $\mathbb{A}^k \subseteq \mathcal{E}$ (levels $k = 1, 2, ..., \infty$ in the inset image, where each $k$-th level below subdivides each pixel in level 0 into $N_k = 2^k \times 2^k$ sub-pixels.) Due to the properties of white Gaussian noise, integrating sub-pixels of the noise defined on $\mathbb{A}^k$ maintains the properties of noise defined on $\mathbb{A}^0$. If we assume there is only one single pixel sample in the domain $(D = 1)$ $A^0 = [0,1] \times [0,1]$, with $\mathbb{A}^k = \{A_1^k, \ldots, A_{N_k}^k\}$ representing the $N_k$ sub-pixels at a finer resolution $k$, the following holds:

$$\sum_{i=1}^{N_k} W(A_i^k) = W(\bigcup_{i=1}^{N_k} A_i^k) = W(A^0). \tag{1}$$

The proposed $\int$-noise refers to the idea that instead of representing a noise value in a discrete point of domain, we rather represent the white noise integral over a pre-specified area. This property is represented in Equation (1) and a more thorough explanation is presented in Appendix B.1. Since we assume that each pixel on the coarsest level $\mathbb{A}^0$ has unit area, the noise variance $\nu_k = \nu(A_i^k)$ at each level is implicitly scaled by the sub-pixel area as $\nu_k = 1/N_k$. While it is impossible to sample the noise $\mathbb{A}^\infty$ in the infinite setting, we show in Section 2.2 that approximating it with a higher-resolution grid is sufficient for a temporally coherent noise transport.

**Conditional white noise sampling.** In practice, after obtaining an *a priori* noise, e.g. from noise inversion techniques in diffusion models, one important aspect is how to construct the $\int$-noise at $\mathbb{A}^k$ from samples defined at $\mathbb{A}^0$. This is fundamentally a conditional probability question: given the value of an entire pixel, what is the distribution of its sub-pixels values? Let $W(\mathbb{A}^k) = \left( W(A_1^k), \ldots, W(A_{N_k}^k) \right)^\top \sim \mathcal{N}(\mathbf{0}, \nu_k \mathbf{I})$ be the $N_k$-dimensional Gaussian random variable representing sub-pixels of a single pixel. Then, the conditional distribution $\left( W(\mathbb{A}^k) | W(A^0) = x \right)$ is

$$\left( W(\mathbb{A}^k) | W(A^0) = x \right) \sim \mathcal{N} \left( \bar{\boldsymbol{\mu}}, \bar{\boldsymbol{\Sigma}} \right), \quad \text{with } \bar{\boldsymbol{\mu}} = \frac{x}{N_k} \mathbf{u}, \bar{\boldsymbol{\Sigma}} = \frac{1}{N_k} \left( \mathbf{I}_{N_k} - \frac{1}{N_k} \mathbf{u} \mathbf{u}^\top \right), \tag{2}$$

where $\mathbf{u} = (1, \ldots, 1)^\top$. By setting $\boldsymbol{U} = \sqrt{N_k} \bar{\boldsymbol{\Sigma}}$, the reparameterization trick gives us a simple way to sample $W(\mathbb{A}^k)$ as

$$(W(\mathbb{A}^k) | W(A^0) = x) = \bar{\boldsymbol{\mu}} + \boldsymbol{U} Z = \frac{x}{N_k} \mathbf{u} + \frac{1}{\sqrt{N_k}} (Z - \langle Z \rangle \mathbf{u}), \quad \text{with } Z \sim (\mathbf{0}, \mathbf{I}), \tag{3}$$

where $\langle Z \rangle$ is the mean of $Z$. Intuitively, in order to conditionally sample the white noise under a pixel of value $x$ at level $k$, one can 1) unconditionally sample a discrete $N_k = 2^k \times 2^k$ Gaussian sample, 2) remove its mean from it, and 3) add the pixel value $x$ (up to a scaling factor). The full derivation of Equations (2) and (3) and the corresponding python code are included in Appendix B.

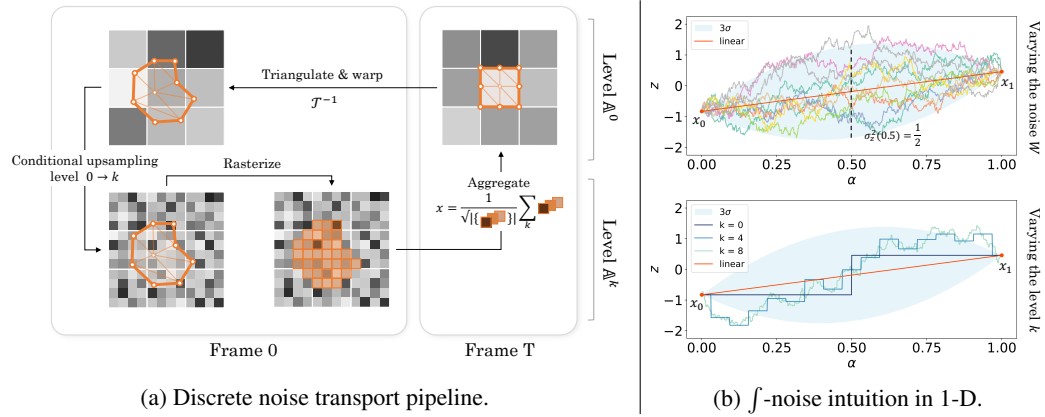

(a) Discrete noise transport pipeline.  (b) $\int$-noise intuition in 1-D.

Figure 2: (a) **The discrete noise transport equation pipeline.** A subdivided pixel contour (top right) is triangulated and traced backwards from frame $T$ to frame 0 (top left). Then the warped triangulated shape is rasterized into a higher resolution approximation of the white noise (bottom). The sub-pixel values are added together, and properly scaled by Equation (5). (b) **1-D toy example.** A pixel slides between two existing pixels whose values $x_0$, $x_1$ are sampled from a Gaussian distribution. Bilinear interpolation creates a sample of lower variance (straight line), whereas $\int$-noise would create samples that follow a Brownian bridge between $x_0$ and $x_1$, maintaining a unit variance.

## 2.2 TEMPORALLY-CORRELATED DISTRIBUTION-PRESERVING NOISE TRANSPORT

In this section, we first introduce our proposed *noise transport equation*, which offers a theoretical way of warping a continuously defined white noise while preserving its characteristics. Then, we present our practical implementation and show that it still retains many theoretical guarantees from the infinite setting. Lastly, we provide a simple 1-D example to support the analysis of how the proposed $\int$-noise balances between satisfying seemingly opposite objectives such as preserving the correlation imposed by interpolation and maintaining the original noise distribution.

**White noise transport.** We first assume that the noise is transported with a diffeomorphic deformation field $\mathcal{T} : E \to E$. This mapping could be represented by an optical flow field between two frames, or a deformation field for image editing. Our goal is to transport a continuous white noise $W$ with $\mathcal{T}$ in a distribution-preserving manner. The resulting noise $\mathcal{T}(W)$ can be expressed as an Itô integral through our *noise transport equation* for any subset $A \subseteq E$ as

$$\mathcal{T}(W)(A) = \int_{\mathbf{x} \in A} \frac{1}{|\nabla \mathcal{T}\left(\mathcal{T}^{-1}(\mathbf{x})\right)|^{\frac{1}{2}}} W(\mathcal{T}^{-1}(\mathbf{x})) \, d\mathbf{x}, \tag{4}$$

where $|\nabla \mathcal{T}|$ is the determinant of the Jacobian of $\mathcal{T}$. Intuitively, Equation (4) warps a non-empty set of the domain with the deformation field $\mathcal{T}^{-1}$, fetching values from the original white noise at the warped domain coordinates. The determinant of the Jacobian is necessary to rescale the samples according to the amount of local stretching that the deformation induces, while also accounting for the variance change required by the white noise definition. A detailed derivation of Equation (4) can be found in the Appendix C.2. In practice, optical flow maps can be non-diffeomorphic due to discontinuities and disocclusions. Appendix C.4 explains our treatment of these cases.

**Discrete Warping.** The noise transport equation (Equation (4)) cannot be solved in practice due the infinite nature of the white noise. Thus, we first compute the higher-resolution discrete $\int$-noise $W(\mathbb{A}^k)$, possibly from an *a priori* sample (Equation (3)). Since the pixel area can undergo a non-linear warping deformation, we subdivide its contour into $s$ smaller segments which are mapped backwards via the reverse deformation field. The warped segments define a polygonal shape that is then triangulated and rasterized over the high-resolution domain $\mathbb{A}^k$. Lastly, the sub-pixels covered by the warped polygon are summed together and normalized, which yields the discrete noise transport for the noise pixel at position $\mathbf{p}$

$$G(\mathbf{p}) = \frac{1}{\sqrt{|\Omega_{\mathbf{p}}|}} \sum_{A_i^k \in \Omega_{\mathbf{p}}} W_k(A_i^k) \,, \tag{5}$$

Figure 3: We visualize the correlation between two $4 \times 4$ noise samples with the warping being a horizontal shift by $\Delta x = 3.6$ pixels. Our $\int$-noise prior preserves the correlation between the two noise samples as well as bilinear interpolation (left), while avoiding self-correlation between pixels in the warped noise (right).

where $W_k = \sqrt{N_k} \cdot W$ is the white noise scaled to unit variance at level $k$, and $\Omega_{\mathbf{p}} \subseteq \mathbb{A}^k$ contains all subpixels at level $k$ that are covered by the warped pixel polygon, with $|\Omega_{\mathbf{p}}|$ representing the cardinality of the set. A detailed algorithm is outlined in Appendix C.4, and illustrated in Figure 2a. Note that the discrete implementation will still preserve independence between neighboring pixels in the warped result. This is because the warped polygons still form a partition of the space, so each sub-pixel in $\mathbb{A}^k$ will only belong to a single warped polygon. More details on the discretization of Equation (4) into Equation (5) can be found in Appendix C.3.

**Toy example in 1-D.** We will demonstrate the properties of our $\int$-noise sampling in a simpler one dimensional setting. Consider a 1-D set of *i.i.d.* random variables indexed by $\mathcal{I} = \{0, \ldots, n\}$ with values represented by $\{x_0, x_1, \ldots, x_n\} \sim \mathcal{N}(0, 1)$, and a mapping function that translates the discrete locations by a constant $\mathcal{T}_{1D}^{-1}(i) = i - \alpha$, where $i \in \mathcal{I}$ and $\alpha \in [0, 1]$. Using a simple linear interpolation to compute the transported values $z_i$ yields

$$z_i = \alpha x_{i-1} + (1 - \alpha)x_i, \quad z_i \sim \mathcal{N}(0, \sigma_z^2), \quad \text{with } \sigma_z^2 = \alpha^2 + (1 - \alpha)^2.$$

The equation above means that the variance of $z_i$ is a quadratic function of $\alpha$ such that $\sigma_z^2 = 1$ for $\alpha \in \{0, 1\}$ and $\sigma_z^2 < 1$ for $\alpha \in (0, 1)$. This shows that the linear interpolation does not preserve the original distribution of the input variables.

However, if we obtain $x_{i-1}$ and $x_i$ from the integral over an underlying high-resolution white noise, the original distribution can be preserved. By employing the $\int$-noise, the pixel's value that is sampled between $x_{i-1}$ and $x_i$ is no longer deterministic. Figure 2b shows the value of $z$ for different higher-resolution samples. Mathematically, the value of $z$ is now a *Brownian bridge* between neighboring $x$ values. In Appendix C.5, we show that the value of $z$ is a conditional probability distribution given by

$$z_i | x_i, x_{i-1} \sim \mathcal{N}(\mu_\infty, \sigma_\infty^2), \quad \text{with } \begin{cases} \mu_\infty = \alpha x_{i-1} + (1 - \alpha)x_i \\ \sigma_\infty^2 = 1 - (\alpha^2 + (1 - \alpha)^2) = 1 - \sigma_z^2 \end{cases} \tag{6}$$

Thus, our continuous noise warping can be interpreted as performing a stochastic process centered around the result of a linear interpolation. This stochastic component precisely compensates the diminished variance induced by the linear interpolation, resulting in a unit variance for any $\alpha$. This is the intuition behind why our $\int$-noise warping method is able to preserve the distribution of the noise sample after warping.

**Long-term temporal coherency.** The method outlined by previous sections only explains how to consistently warp a noise sample with a single deformation field. To apply this idea to a full sequence of frames, there are two possible solutions. The first one is to warp the noise frame by frame, such that the noise $G_n$ of the $n$-th frame is computed as $G_n = \mathcal{T}_{(n-1) \to n}(G_{n-1})$. However, the higher-resolution noises computed at each frame are no longer coherent with each other in this case. A better solution is to use the accumulated deformation field to warp back to the first sample, where the high resolution representation is always the same, i.e., $G_n = \mathcal{T}_{0 \to n}(G_0)$. This produces noises that are more coherent over long periods of time.

## 3 EXPERIMENTS AND RESULTS

### 3.1 VALIDATING $\int$-NOISE PRIOR

Our proposed noise prior simultaneously achieves two seemingly competing objectives: maximizing the correct correlation between the warped and the original sample, and maintaining the indepen-

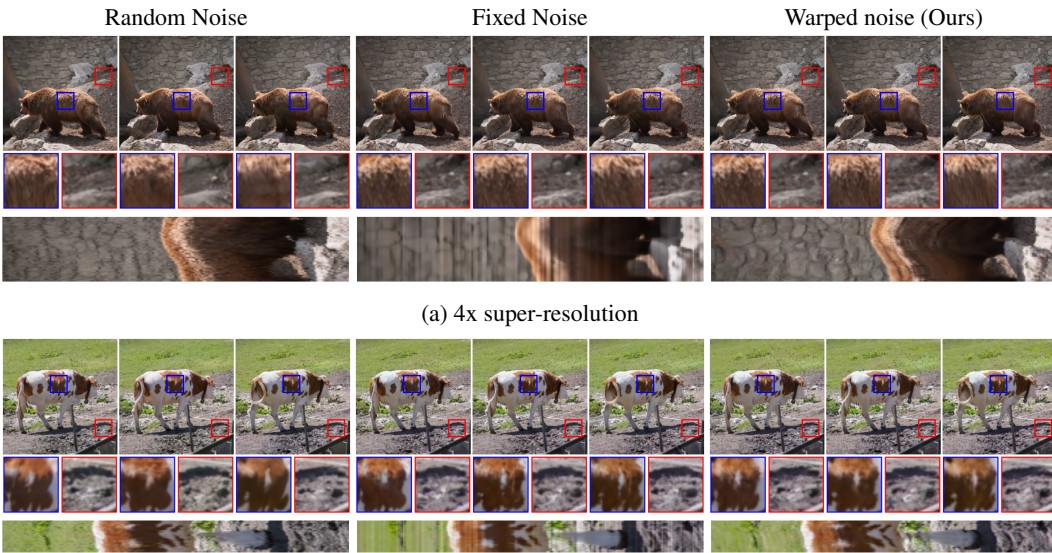

(a) 4x super-resolution

(b) JPEG compression restoration

Figure 4: Qualitative comparison of our noise warping method with baselines for video restoration tasks using image models by I²SB: consecutive frames comparison (top), $x$-$t$ slice (bottom).

dence of pixels within each sample. These two aspects can be visualized using cross-covariance and covariance matrices of two noise samples $G_0$ and $G_1$ respectively. As shown in Figure 3, our $\int$-noise prior achieves the same noise cross-correlation as the bilinear interpolation, while concurrently maintaining the same level of independence between pixels in the second noise sample $G_1$ as methods such as resampling (*"random"*) or reusing the previous noise (*"fixed"*).

## 3.2  APPLICATIONS IN DIFFUSION-BASED TASKS

We validate our noise prior on four tasks: realistic appearance transfer with SDEdit (Meng et al., 2022); video restoration and super-resolution with I²SB (Liu et al., 2023a); pose-to-person video generation (Bhunia et al., 2022), and fluid simulation super-resolution. While we show qualitative examples of these applications in this paper, we urge the reader to check out the supplementary videos on our project webpage for better visual comparisons.

**Baselines.** In many of our applications, we lift models trained on images to work on videos. Without any particular treatment, the default *Random Noise* prior uses different and independent noise samples for each frame. Alternatively, many approaches employ a *Fixed Noise* prior to reduce flickering artifacts, which reuses a same fixed set of noise samples for all frames. We compare the proposed $\int$-noise mainly against these two baselines, but more comparisons can be found in Appendix D.

**Photorealistic appearance transfer for videos.** We lift the stroke-based editing capabilities of SDEdit (Meng et al., 2022) to the temporal domain. We showcase results on the LSUN Bedroom dataset (Yu et al., 2015) by replacing the stroke paintings with simple renders of a synthetic bedroom scene to ensure that the inputs are temporally coherent. SDEdit then corrupts these frames and denoise them into realistic images. Figure 1a shows that our $\int$-noise prior significantly improves the temporal coherency of the denoised sequence by avoiding unnatural sticking artifacts. While our method also works without it, we use cross-frame attention in this specific application to better showcase the difference between the different noise priors.

**Video restoration and super-resolution.** In I²SB, Liu et al. (2023a) use Schrödinger bridges to diffuse between two distributions without going through the normal Gaussian distribution. The authors show examples for JPEG compression restoration and $4\times$ image super-resolution. When using their model on a per-frame basis for videos, the resulting images suffer from artifacts due to the choice of noise. By leveraging the motion information in the corrupted sequence to warp the

| Random noise | Fixed noise | ∫-noise (ours) |

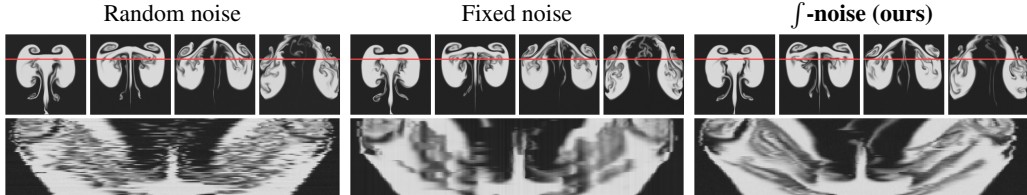

Figure 5: **Fluid $4\times$ super-resolution.** As shown in $x$-$t$ slices (bottom row), *Random Noise* creates incoherent details (noise in the slice) while *Fixed Noise* suffers from sticking artifacts (vertical lines in the slice). Our ∫-noise moves the fluid in a smoother way.

noise, we can noticeably reduce these temporal aberrations, as shown in the $x$-$t$ plots of Figure 4. Quantitative comparisons in Table 1 substantiate these observations.

**Pose-to-person video.** Person Image Diffusion Model (PIDM) (Bhunia et al., 2022) is a non-latent diffusion-based method that takes an image of a person and a pose, and generates an image of the same person with the input pose. When applying the model in a frame-by-frame fashion on a sequence of poses, it either creates a substantial amount of unnatural flickering when using the Random Noise prior, or produces obvious cloth texture-sticking artifacts when combined with fixed noise (see blue squares in Figure 1b). We estimate a rough motion of the entire body from the pose sequence and show that this information can be utilized with our noise warping method to alleviate the texture sticking issue. Figure 1b shows how ∫-noise translates the cloth textures along with the person's movements more naturally. Interestingly, this does not fully agree with the quantitative evaluation, and we further analyse this discrepancy in Appendix D.3.

**Fluid simulation super-resolution.** Finally, we apply our ∫-noise prior to a challenging task of fluid super-resolution. Fluid simulations contains large motions and deformations which makes artifacts from standard noise priors much more visible. Large deformations also means that simpler warping methods like bilinear interpolation will fail miserably. We train a non-latent unconditional diffusion model from scratch on a 2D fluid simulation dataset generated with the fluid solver from Tang et al. (2021). The $x$-$t$ plots in Figure 5 show that warping the noise along with the fluid density creates smoother transitions between frames.

**Quantitative evaluation.** We quantitatively evaluate our noise prior against the baselines presented above, as well as some traditional interpolation methods in Table (1). We further add the noise priors from Ge et al. (2023) and Chen et al. (2023) to our evaluation. Following Geyer et al. (2023); Ceylan et al. (2023); Lai et al. (2018), we use the warping error as a metric for temporal coherency. Realism and visual quality of the results are assessed with FID Heusel et al. (2017), Improved Precision Kynkäänniemi et al. (2019) and/or LPIPS Zhang et al. (2018) whenever it makes sense. Additionally, we report the runtime of each method in Appendix C.7. Our method is computationally less efficient than the baselines in Table (1), but remains comparable to DDIM inversion.

| Method | Appearance Transfer | | | Video SR ($4\times$) | | JPEG restore | | Pose-to-Person |
| --- | --- | --- | --- | --- | --- | --- | --- | --- |
| | warp $_{(\times 10^{-3})}\downarrow$ | FID $\downarrow$ | Precision $\uparrow$ | warp $\downarrow$ | LPIPS $\downarrow$ | warp $\downarrow$ | LPIPS $\downarrow$ | warp $\downarrow$ |
| Random | 10.00 | 74.75 | 0.719 | 8.91 | 0.192 | 8.28 | 0.163 | 34.22 |
| Fixed | 4.35 ● | 93.56 | 0.644 | 7.97 ● | 0.179 | 7.54 | 0.163 | 2.26 ● |
| PYoCo (mixed) | 7.26 | 81.60 | 0.674 | 8.48 | 0.190 | 7.77 | 0.166 | 20.92 |
| PYoCo (prog.) | 4.90 | 84.63 | 0.667 | 8.10 | 0.190 | 7.48 | 0.163 | 11.97 |
| Control-A-Video | 5.09 | 90.82 | 0.649 | 7.98 ● | 0.192 | 7.73 | 0.164 | 6.45 |
| Bilinear | 3.15 ● | 143.24 | 0.201 | 21.47 | 0.590 | **5.26** ● | 0.431 | **2.13** ● |
| Bicubic | 4.95 | 149.85 | 0.212 | 13.02 | 0.490 | 5.74 ● | 0.372 | 2.40 ● |
| Nearest | 15.10 | 154.73 | 0.344 | 14.30 | 0.329 | 7.91 | 0.213 | 9.21 |
| ∫-noise (ours) | **2.50** ● | 92.63 | 0.661 | **6.49** ● | 0.196 | 5.98 ● | 0.165 | 2.92 |

Table 1: Quantitative evaluation of our method on the different applications. Our method outperforms existing noise priors in terms of temporal coherency, while being competitive to traditional interpolation methods. Additionally, our noise prior is able to generate results with visual quality on par with standard Gaussian priors thanks to its distribution-preserving properties.

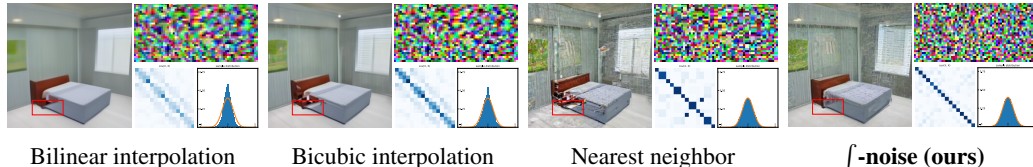

| Bilinear interpolation | Bicubic interpolation | Nearest neighbor | ∫-noise (ours) |

Figure 6: We compare our ∫-noise against standard interpolation methods on the task of photorealistic appearance transfer for videos. For each method, we visualize a frame from the denoised result, a patch from one of the noise we used, the covariance matrix of the noise and the overall noise distribution. The covariance matrix is computed on a batch of 100 noises on a small $4 \times 4$ patch inside the visualized noise. The standard normal distribution is plotted in orange.

## 4 ABLATION STUDIES AND DISCUSSIONS

**Standard interpolation vs ∫-noise.** Figure 6 shows that bicubic and bilinear interpolation schemes for noise warping lead to blurry results, because the warped noise has less variance (histogram insets). While nearest neighbor interpolation preserves the distribution to a certain extent, it contains duplicating artifacts in regions of stretching (noise inset), which induces self-correlation between pixels. Our proposed method is able to retain high-frequency details in the result, and ensures complete independence between the pixels of the noise samples.

**The noise upsampling factor $k$.** The main parameter of our noise warping algorithm is the choice of the upsample factor $k$. A smaller factor $k$ means the warped polygons are rasterized on larger sub-pixels, increasing the chances that the polygon covers none of the sub-pixels. In this case, the resulting pixel value after warping is undefined. We test our method on the fluid example, where large deformations are present. The inset plot shows the ratio of undefined pixels in the warped noise for different values of $k$ and different 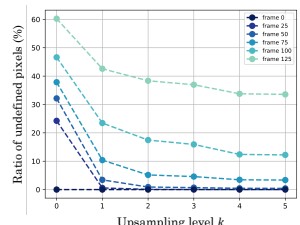 frames. The further the frame, the more undefined pixels appear due to the extreme warping the pixel polygons go through. Overall, sub-sampling the noise to level $k = 1$ already largely reduces the amount of unrasterized pixel polygons. We found setting $k = 3$ is sufficient for most cases. For very long sequences, we devise a resampling strategy detailed in Appendix C.4.

**Noise warping in latent diffusion models.** Our experiments have shown that noise warping has limited impact on temporal coherency in latent diffusion models (LDM) (Rombach et al., 2022). This is because the noise used in latent diffusion models has a lower resolution, controlling mostly the compositional aspects and low frequency structures of the generated image. This not only limits the amount of motion that can be transferred to it but also offloads the responsibility of generating the final high-frequency details to the decoder part of the image autoencoder. In Appendix E, we provide an extensive set of tests that highlight the impact of using the warped noise in LDMs.

## 5 RELATED WORK

**Diffusion models** (Sohl-Dickstein et al., 2015; Ho et al., 2020) represent the state the art for generative models, surpassing previous techniques in terms of sampling quality and mode coverage (Dhariwal & Nichol, 2021; Xiao et al., 2022). Diffusion models train a neural network to reverse a diffusion process, representing the data manifold through a Gaussian distribution. The set of sampled noises, along with integrator parameters, can be interpreted as the latent space of generative diffusion models (Huberman-Spiegelglas et al., 2023). Since their original conception, diffusion models have been employed in a plethora of tasks such as text-to-image (Rombach et al., 2022; Nichol et al., 2022; Ramesh et al., 2022; Balaji et al., 2022); text-to-video (Ho et al., 2022a;b; Singer et al., 2022; Villegas et al., 2022), image-to-image Parmar et al. (2023) and image-to-video Ni et al. (2023) generation; image (Meng et al., 2022; Valevski et al., 2022) and video (Yang et al., 2023; Chen et al., 2023; Ceylan et al., 2023) editing; and image (Saharia et al., 2022; Lugmayr et al., 2022; Chung et al., 2022) and video inpainting Liew et al. (2023).

**Diffusion-based image editing through noise inversion.** It is common practice for image editing methods (Tumanyan et al., 2022; Ceylan et al., 2023; Mokady et al., 2022; Parmar et al., 2023;

Huberman-Spiegelglas et al., 2023; Wu & De la Torre, 2022; Yang et al., 2023; Geyer et al., 2023) to invert the denoising process, recovering the noise that reconstructs an image under a textual condition. The inverted noise partially encodes the image structural composition, and through its recombination with a new textual prompt during the denoising process, edited images can be efficiently generated. This inversion process can either reconstruct a single (DDIM) (Hertz et al., 2022; Mokady et al., 2022; Song et al., 2021) or multiple (DDPM) (Huberman-Spiegelglas et al., 2023; Wu & De la Torre, 2022) noise samples per image. Inverting the noise can be inaccurate and exact inversion can be obtained by coupled transformations (Wallace et al., 2022) or by bi-direction integration approximation (Zhang et al., 2023). To further increase spatial fidelity to the original image when implementing prompt-to-prompt edits, some approaches advocate for injecting cross-frame attention (Hertz et al., 2022; Parmar et al., 2023) and features maps (Tumanyan et al., 2022) from the source image during the denoising process. Noise inversion techniques were also used to perform image-to-image style transfer (Zhang et al., 2022; Ruta et al., 2023), combined with a noise regularization technique (Parmar et al., 2023) and improved by pivotal tuning and null-text optimization (Mokady et al., 2022).

**Diffusion-based video generation and editing.** Similar to image editing, many video editing applications rely on a text-to-image models to invert the noise for each individual frame. Temporal coherency is a central challenge for these applications. Injecting cross-attention features (Ceylan et al., 2023; Geyer et al., 2023; Yang et al., 2023; Khachatryan et al., 2023) from an anchor frame is a possible solution to ensure consistent edits. Recent techniques rely on modifying the noise or the latent space of diffusion models to achieve better temporal coherency: Ni et al. (2023) warp the latent features from a diffusion model of a image-to-video pipeline; Blattmann et al. (2023) align the latent codes of a pre-trained image diffusion model during fine-tuning; Khachatryan et al. (2023) apply pre-specified translation vectors to the inverted noise from an anchor frame; Ge et al. (2023) construct correlated noise samples to train a neural network for generating videos from text; Chen et al. (2023) use a residual noise sampling that spatially preserve noise samples representing image pixels not moving in the video sequence. Alternatively, Layered Neural Atlases (Kasten et al., 2021) decomposes the video into a canonical editing space (Couairon et al., 2023; Chai et al., 2023).

**Image and video editing through fine-tuning.** An alternative approach to deal with issues arising from image and video editing is to fine-tune neural networks on top of diffusion models. Control-net (Zhang & Agrawala, 2023) learns task-specific conditions (depth and normal maps, edges, poses) that are useful for preserving structures while editing images. For editing videos, several approaches fine-tune neural network with temporal-attention modules (Liew et al., 2023; Liu et al., 2023b; Shin et al., 2023; Zhao et al., 2023; Wu et al., 2022) to ensure temporally coherent outputs. Alternatively, embedding (Chu et al., 2023) or disentangling (Guo et al., 2023) motion and the content from videos is also an effective approach to control consistency.

## 6 Limitations & Conclusion

In this paper, we proposed a novel temporally-correlated noise prior. Our $\int$-noise representation reinterprets noise samples as the integral of a higher resolution white noise field, allowing variance-preserving sampling operations that are able to continuously interpolate a noise field. Moreover, we derived a novel noise transport equation, which accounts not only for the deformation of the underlying pixel shapes but also for the necessary variance rescaling. The proposed method creates temporally-correlated and distribution-preserving noise samples that are useful for a variety of tasks.

Our method comes with some limitations. While more accurate, the proposed noise warping algorithm is computationally more inefficient than simpler techniques (Ge et al., 2023; Chen et al., 2023) for correlation preservation. Moreover, the underlying assumption we made is that a more temporally-correlated noise prior would result in better temporal coherency in video diffusion tasks. This assumption is not always guaranteed. The degree to which this holds depends on the training data and the pipeline of the diffusion model: while methods such as SDEdit give enough freedom to the noise for it to have a visible impact on the denoised result, more constrained pipelines can be much more oblivious to the choice of the noise prior. For future work, we believe that our noise warping algorithm can be further used to train neural networks that generate video inputs. Extending the noise prior to latent diffusion models can be another interesting direction. And that, dear reader, is how we warped your noise.

## 7 REPRODUCIBILITY STATEMENT

1. For applications using existing methods, we use the checkpoints provided by the authors of the methods.
2. Proofs are provided in Appendix B and C.
3. Pseudocode are provided in Appendix B.3 and C.4.
4. Extra experimental results are provided in Appendix D.
5. Detailed analysis of our method in latent diffusion models is provided in Appendix E.

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

## A    LIST OF SYMBOLS

| | |
|---:|---|
| $D, (i,j)$ | Noise dimensions, pixel coordinates |
| $G, G_n$ | Discrete Gaussian noise sample (at frame $n$) |
| $G(i,j), G(\mathbf{p})$ | Random variable obtained by evaluating $G$ at pixel coordinates $(i,j)$ or $\mathbf{p}$ |
| $X_{i,j}, Z$ | Gaussian distributed variables |
| $\mathcal{B}(E)$ | Borel $\sigma$-algebra on $E$, i.e. collection of all Borel sets on $E$ |
| $\nu$ | Standard Lebesgue measure, corresponds to the area measure in $\mathbb{R}^2$ |
| $(E, \mathcal{E}, \nu)$ | $\sigma$-finite measure space on $E$, with Borel $\sigma$-algebra $\mathcal{E} = \mathcal{B}(E)$ and measure $\nu$ |
| $W$ | Continuous white Gaussian noise as a set function |
| $W(A)$ | White noise defined over $A \in \mathcal{B}(E)$, $W(A)$ is a Gaussian random variable |
| $\mathbb{A}^0, \mathbb{A}^k, \mathbb{A}^\infty$ | Partitions of the domain $E$ into regular grids at different resolutions, $\mathbb{A}^k \subset \mathcal{E}$ |
| $N_k, \nu_k$ | Number of per-pixel sub-pixel samples and noise variance for a $k$-th level |
| $x$ | *a priori* noise sample |
| $\bar{\boldsymbol{\mu}}, \bar{\boldsymbol{\Sigma}}$ | Discrete multivariable mean and variance |
| $\langle \cdot \rangle$ | Mean operator |
| $\mathcal{T}$ | Deformation field used in warping |
| $\mathcal{T}(W)$ | Warping $W$ with $\mathcal{T}$ using the noise transport equation, $\mathcal{T}(W)$ is a white noise |
| $\Omega_{\mathbf{p}}$ | A subset of $\mathbb{A}^k$, i.e. $\Omega_{\mathbf{p}} \subseteq \mathbb{A}^k$ |
| $k, s$ | Noise up-sampling level, piecewise discretization level of pixel polygons |

Table 2: List of mathematical symbols used in the main text.

## B    CONDITIONAL WHITE NOISE SAMPLING

An important step in our pipeline is to lift a given initial discrete noise sample to a continuously defined white noise function. In Section 2.1, we introduced a method for conditionally sampling a white noise at any finer resolution given a discrete sample. In this section we outline additional important white noise properties; for a more rigorous white noise definition please refer to (Walsh, 2006; Dalang & Mountford, 1999). We also include a proof of Equation 2 alongside with the pseudo-code for conditional sampling.

### B.1    WHITE NOISE

Let $(\Omega, \mathcal{F}, P)$ be a complete probability space and let $(E, \mathcal{E}, \nu)$ be a $\sigma$-finite measure space. White noise on $E$ based on $\nu$ is a random set function

$$A \mapsto W(A), \tag{7}$$

defined for $A \in \mathcal{E}$ with $\nu(A) < \infty$, with values in $L^2(\Omega, \mathcal{F}, P)$, such that:

1. $W(A)$ is a Gaussian random variable with mean 0 and variance $\nu(A)$;

2. if $A$ and $B$ are disjoint, then $W(A)$ and $W(B)$ are independent and

$$W(A \cup B) = W(A) + W(B).$$

In our setting, we usually consider the continuous white noise to be defined on $E \subseteq \mathbb{R}^2$. A useful fact about the covariance of $W(A)$ and $W(B)$ is that it can be computed from the definition:

$$\begin{aligned} E(W(A)W(B)) &= E((W(A \backslash B) + W(A \cap B))(W(B \backslash A) + W(B \cap A))) \\ &= E\left(W(A \cap B)^2\right) \\ &= \nu(A \cap B). \end{aligned} \tag{8}$$

## B.2 CONDITIONAL PROBABILITY DERIVATION

**Setup.** To simplify the derivation and without loss of generality, let us consider a single discrete pixel associated with a Gaussian multivariable $\mathbf{X} \sim \mathcal{N}(\mathbf{0}_1, \mathbf{I}_1)$, where $\mathbf{0}_1$ and $\mathbf{I}_1$ represent the zero and identity square matrices of dimensionality 1. In our noise formulation, the pixel does not represent a point, but it rather defines a constant value in a unit square $A = [0,1]^2$. Our goal is to subdivide the pixel area $A$ into $N \times N$ sub-pixels, and sample them conditionally given an *a priori* sample $\mathbf{X} = x$. To do so, we start by partitioning $A$ into $N^2$ sub-patches $\{B_{k,l} = [\frac{k-1}{N}, \frac{k}{N}] \times [\frac{l-1}{N}, \frac{l}{N}]\}_{(k,l) \in [1,N]^2}$, such that

$$\nu(B_{k,l}) = \frac{1}{N^2}, \quad W(B_{k,l}) \sim \mathcal{N}(0, \frac{1}{N^2}), \quad \forall\, k,l \in [1,\ldots,N]^2. \tag{9}$$

**Proof of Equation (2).** Let $\mathbf{Y} = (W(B_{1,1}), ..., W(B_{N,N}))^\top$ be the random vector associated with all the sub-patches. $\mathbf{Y}$ is a multivariate Gaussian random vector thanks to the independence of $\{W(B_{k,l})\}_{k,l}$, and we have $\mathbf{Y} \sim \mathcal{N}(\boldsymbol{\mu_Y} = \mathbf{0}_{N^2}, \boldsymbol{\Sigma_Y} = \frac{1}{N^2}\mathbf{I}_{N^2})$. Now, consider the vector $\mathbf{Z} = (\mathbf{Y}, \mathbf{X})^\top$. From Equation (1), we have that $\mathbf{X} := W(A) = W\left(\bigcup_{k,l} B_{k,l}\right) = \sum_{k,l} W(B_{k,l})$, i.e. $\mathbf{X}$ is a linear combination of $\mathbf{Y}$. As a result, any linear combination of elements in $\mathbf{Z}$ is also a linear combination of elements in $\mathbf{Y}$, thus a Gaussian variable. Therefore, $\mathbf{Z}$ is a multivariate Gaussian variable, and:

$$\mathbf{Z} \sim (\boldsymbol{\mu}, \boldsymbol{\Sigma}), \quad \text{with } \boldsymbol{\mu} = \begin{bmatrix} \boldsymbol{\mu_Y} \\ \boldsymbol{\mu_X} \end{bmatrix}, \; \boldsymbol{\Sigma} = \begin{bmatrix} \mathbf{C_{(Y,Y)}} & \mathbf{C_{(Y,X)}} \\ \mathbf{C_{(X,Y)}} & \mathbf{C_{(X,X)}} \end{bmatrix}, \tag{10}$$

where $\mathbf{C}$ constructs the covariance matrix for two given variables. Given the white noise definitions in Appendix B.1, the covariance between two individual Gaussian random variables defined on $A$ (pixel) and $B_{k,l}$ (sub-pixel) is their intersected area $\nu(B_{k,l} \cap A) = \nu(B_{k,l}) = \frac{1}{N^2}$. This yields the following covariance matrices:

$$\mathbf{C_{(Y,Y)}} = \frac{1}{N^2}\mathbf{I}_{N^2}, \quad \mathbf{C_{(Y,X)}} = \frac{1}{N^2}\mathbf{u}, \quad \mathbf{C_{(X,Y)}} = \mathbf{C_{(Y,X)}}^\top, \quad \mathbf{C_{(X,X)}} = \mathbf{I}_1, \tag{11}$$

where $\mathbf{u} = (1, ..., 1)^\top \in \mathbb{R}^{N^2}$. Since $\mathbf{Z}$ is a Gaussian vector, the conditional probability of $\mathbf{Y}|\mathbf{X} = x$ is also a Gaussian, and its sampling is given by (Holt & Nguyen, 2023)

$$\mathbf{Y}|\mathbf{X} \sim \mathcal{N}\left(\boldsymbol{\mu_{(Y|X)}}, \boldsymbol{\Sigma_{(Y|X)}}\right), \quad \text{where} \begin{cases} \boldsymbol{\mu_{(Y|X)}} = \boldsymbol{\mu_Y} + \mathbf{C_{(Y,X)}}\, \mathbf{C_{(X,X)}}^{-1}\, (\mathbf{X} - \boldsymbol{\mu_X}), \\ \boldsymbol{\Sigma_{(Y|X)}} = \mathbf{C_{(Y,Y)}} - \mathbf{C_{(Y,X)}}\, (\mathbf{C_{(X,X)}})^{-1}\, \mathbf{C_{(X,Y)}}. \end{cases} \tag{12}$$

The terms above further simplify using Equation (11) as

$$\boldsymbol{\mu_{(Y|X)}} = \boldsymbol{\mu_Y} + \mathbf{C_{(Y,X)}}\, (\mathbf{C_{(X,X)}})^{-1}\, (\mathbf{X} - \boldsymbol{\mu_X}) = \mathbf{0}_{N^2} + \frac{1}{N^2}\mathbf{u}(\mathbf{X} - \mathbf{0}_1) = \frac{x}{N^2}\mathbf{u},$$

$$\boldsymbol{\Sigma_{(Y|X)}} = \mathbf{C_{(Y,Y)}} - \mathbf{C_{(Y,X)}}\, \mathbf{C_{(X,Y)}} = \frac{1}{N^2}\mathbf{I}_{N^2} - \frac{1}{N^2}\mathbf{u}(\frac{1}{N^2}\mathbf{u})^\top = \frac{1}{N^2}I_{N^2} - \frac{1}{N^4}\mathbf{u}\mathbf{u}^\top. \tag{13}$$

The last missing piece to derive Equation (2) comes by setting $UU^\top = \boldsymbol{\Sigma_{(Y|X)}}$, with $U = \frac{1}{N}\left(\mathbf{I}_{N^2} - \frac{1}{N^2}\mathbf{u}\mathbf{u}^\top\right)$. These definitions can be verified using the following identity:

$$(\mathbf{u}\mathbf{u}^\top)^2 = \mathbf{u}\mathbf{u}^\top\mathbf{u}\mathbf{u}^\top = (\mathbf{u}^\top\mathbf{u})\mathbf{u}\mathbf{u}^\top = N^2\mathbf{u}\mathbf{u}^\top. \tag{14}$$

$\square$

## B.3 A PRIORI SAMPLING IN PYTHON

**Unit variance scaling.** Note that in the previous derivation, the upsampled noise represented by the new pixels $(B_{k,l})$ has a variance of $1/N^2$. Sometimes, it can be more practical to generate higher-resolution discrete noise with unit variance. This is easily fixed by multiplying the Equation (3) by $N$, resulting in the upsampling function

$$\texttt{upsample}_\infty(x, N) := \frac{x}{N}\mathbf{u} + (Z - \langle Z \rangle\mathbf{u}), \quad Z \sim (\mathbf{0}_{N^2}, \mathbf{I}_{N^2}), \quad \langle Z \rangle = \frac{1}{N^2}\mathbf{u}^\top Z \tag{15}$$

Algorithm (1) illustrates in Python a procedure that upsamples a $H \times W$ noise sample by a factor $N$ scaling it to unit variance.

---

**Algorithm 1** Conditional White Noise Sampling

---

```python
def upsample_noise(X, N):
    b, c, h, w = X.shape
    Z = torch.randn(b, c, N*h, N*w)
    Z_mean = Z.unfold(2, N, N).unfold(3, N, N).mean((4, 5))
    Z_mean = F.interpolate(Z_mean, scale_factor=N, mode='nearest')
    X = F.interpolate(X, scale_factor=N, mode='nearest')
    return X / N + Z - Z_mean
```

---

## C  DISTRIBUTION-PRESERVING NOISE WARPING

The distribution-preserving noise warping method proposed in Section 2.2 involves warping the continuous white noise with a diffeomorphic deformation field. In this section we mathematically derive the noise transport equation (Equation 4).

### C.1  BROWNIAN MOTION AND ITÔ INTEGRAL

**White noise and Brownian motion.** An alternative definition of white noise $\{W(\mathbf{x})\}_{\mathbf{x} \in E}$ is through the *distributional derivative* of a Brownian motion $\{B(\mathbf{x})\}_{\mathbf{x} \in E}$ (also called *Brownian sheet* for dimension $\geq 2$) as $W(\mathbf{x})d\mathbf{x} = dB(\mathbf{x})$.

**Itô integral.** As the Itô integral $\int_A \phi(\mathbf{x})dB(\mathbf{x})$ of a deterministic function $\phi$ in $L^2$ is always Gaussian. The variance is given by:

$$\mathbb{E}\left( \int \phi(\mathbf{x})dB(\mathbf{x}) \right)^2 = \int \phi(\mathbf{x})^2 d\mathbf{x} = \|\phi\|_2^2. \tag{16}$$

From Equation (16) we can relate back to the first definition of white noise by setting $\phi = \mathbf{1}$. Indeed,

$$W(A) = \int_{\mathbf{x} \in A} \mathbf{1}dB(\mathbf{x}) \tag{17}$$

is a Gaussian variable of variance $\int_{\mathbf{x} \in A} \phi(\mathbf{x})^2 d\mathbf{x} = \int_{\mathbf{x} \in A} 1 d\mathbf{x} = \nu(A)$.

### C.2  NOISE TRANSPORT EQUATION DERIVATION

**Warping.** The differential form of the transport equation for pointwise pixel values $\rho(\mathbf{x})$ of an arbitrary image is given by

$$\frac{\partial \rho(\mathbf{x})}{\partial t} = -\nabla \cdot (\rho(\mathbf{x})\,\mathbf{v}(\mathbf{x})), \tag{18}$$

where $\mathbf{v}(\mathbf{x})$ is a velocity field. The mapping $\mathcal{T}$ is defined by integrating the set of positions $\mathbf{x}$ with the velocity field $\mathbf{v}(\mathbf{x})$. More specifically, $\mathcal{T} : E \to E$ is a diffeomorphism from our measure space to itself: $\mathcal{T}$ is differentiable, structure-preserving, and its inverse $\mathcal{T}^{-1}$ is defined for all $\mathbf{x}$. The transport equation can be solved by the method of characteristics mapping (Nabizadeh et al., 2022) to generate the warped field $\tilde{\rho}(\mathbf{x})$ by

$$\tilde{\rho}(\mathbf{x}) = \rho(\mathcal{T}^{-1}(\mathbf{x})). \tag{19}$$

However, this method only considers pointwise pixel values, whereas our definition of white noise requires the transport in continuously defined pixel areas. In 2-D this amounts to regularizing pointwise values by the area that they represent, while simultaneously tracking the distortions resulting from the mapping application. This notion is also common in exterior calculus: 0-forms (points) to

2-forms (faces) are connected through the dual operator (Crane, 2013). We employ a simpler finite volume method to represent $\rho(\Delta\mathbf{x})$ as a cell-centered area-averaged integration

$$\rho(\Delta\mathbf{x}) = \frac{1}{\nu(\Delta\mathbf{x})} \int_{\mathbf{x}\in\Delta\mathbf{x}} \rho(\mathbf{x}) \, d\mathbf{x}. \tag{20}$$

Note that the resulting $\rho(\Delta\mathbf{x})$ can still be considered a pointwise quantity, since it is the integrated value over a pre-specified area regularized by the area size. Applying Equation (20) to $\mathcal{T}^{-1}(\Delta\mathbf{x})$, the warped cell-centered area quantity $\tilde{\rho}(\Delta\mathbf{x})$ is

$$\tilde{\rho}(\Delta\mathbf{x}) = \rho(\mathcal{T}^{-1}(\Delta\mathbf{x})) = \frac{1}{\nu(\mathcal{T}^{-1}(\Delta\mathbf{x}))} \int_{\mathbf{x}\in\mathcal{T}^{-1}(\Delta\mathbf{x})} \rho(\mathbf{x}) \, d\mathbf{x}. \tag{21}$$

If ones naively replaces $\rho$ by a white noise distribution $W$, the warped noise $\widetilde{W}$ follows

$$\widetilde{W}(\Delta\mathbf{x}) = \frac{1}{\nu(\mathcal{T}^{-1}(\Delta\mathbf{x}))} \int_{\mathbf{x}\in\mathcal{T}^{-1}(\Delta\mathbf{x})} W(\mathbf{x}) \, d\mathbf{x}. \tag{22}$$

$\widetilde{W}(\Delta\mathbf{x})$ is a Gaussian random variable, since it is an Itô integral. Its variance is given by evaluating Equation (16) as

$$\sigma^2\left(\widetilde{W}(\Delta\mathbf{x})\right) = \frac{1}{\nu(\mathcal{T}^{-1}(\Delta\mathbf{x}))^2} \int_{\mathbf{x}\in\mathcal{T}^{-1}(\Delta\mathbf{x})} 1^2 \, d\mathbf{x} = \frac{1}{\nu(\mathcal{T}^{-1}(\Delta\mathbf{x}))}. \tag{23}$$

The variance of the integrated white noise above is not correct, as it should be proportional to the integrated area, in accordance with the definitions in Appendix B.1. This shows that a continuous Gaussian white noise loses its distribution when we transport it with standard transport equations. In Section 2.2, we introduced an alternative to the standard transport equations which accounts for variance preservation. We referred to it as the *noise transport equation*

$$\widetilde{W}(A) = \int_{\mathbf{x}\in A} \frac{1}{|\nabla\mathcal{T}\left(\mathcal{T}^{-1}(\mathbf{x})\right)|^{\frac{1}{2}}} W(\mathcal{T}^{-1}(\mathbf{x})) \, d\mathbf{x}, \tag{24}$$

defined for any $A \subseteq E$. This equation correctly accounts for a variance rescaling term, which ensures that the integrated white noise is in conformity with its defining properties. We prove that Equation (24) is the correct formulation that produces a continuously defined noise representation that further satisfies

1. $\widetilde{W}$ is a white noise as defined in Section B.1;

2. $\widetilde{W}$ is a warping of $W$ by $\mathcal{T}$, i.e. it satisfies a relation similar to Equation (19). Specifically,

$$\frac{\widetilde{W}(\Delta\mathbf{x})}{\nu(\Delta\mathbf{x})^{\frac{1}{2}}} = \frac{W(\mathcal{T}^{-1}(\Delta\mathbf{x}))}{\nu(\mathcal{T}^{-1}(\Delta\mathbf{x}))^{\frac{1}{2}}} \tag{25}$$

**Proof 1.** To show that $\widetilde{W}$ is indeed a white noise, we first show that for $A$ s.t. $\nu(A) < \infty$, we have $\widetilde{W}(A) \sim \mathcal{N}(0, \nu(A))$. We proceed to a change of variable $\mathbf{y} = \mathcal{T}^{-1}(\mathbf{x})$, i.e. $|\nabla\mathcal{T}(\mathbf{y})|d\mathbf{y} = d\mathbf{x}$. Thus,

$$
\begin{aligned}
\widetilde{W}(A) &= \int_{\mathbf{y}\in\mathcal{T}^{-1}(A)} \frac{1}{|\nabla\mathcal{T}(\mathbf{y})|^{\frac{1}{2}}} W(\mathbf{y})|\nabla\mathcal{T}(\mathbf{y})|d\mathbf{y} \\
&= \int_{\mathbf{y}\in\mathcal{T}^{-1}(A)} |\nabla\mathcal{T}(\mathbf{y})|^{\frac{1}{2}} W(\mathbf{y})d\mathbf{y} \\
&= \int_{\mathbf{y}\in\mathcal{T}^{-1}(A)} |\nabla\mathcal{T}(\mathbf{y})|^{\frac{1}{2}} dB(\mathbf{y}).
\end{aligned}
\tag{26}
$$

Again, this Itô integral is a Gaussian variable and its variance is given by:

$$
\begin{aligned}
\sigma^2 &= \int_{\mathbf{y} \in \mathcal{T}^{-1}(A)} \left( |\nabla \mathcal{T}(\mathbf{y})|^{\frac{1}{2}} \right)^2 d\mathbf{y} \\
&= \int_{\mathbf{y} \in \mathcal{T}^{-1}(A)} |\nabla \mathcal{T}(\mathbf{y})| \, d\mathbf{y} \\
&= \int_{\mathbf{x} \in A} d\mathbf{x} = \nu(A),
\end{aligned}
\tag{27}
$$

where we used the change of variable the other way round. It is not difficult to verify that $\widetilde{W}$ also satisfies the second condition on disjoint sets, thanks to the linearity of the integral operator and the bijectivity of $\mathcal{T}$. $\qquad\square$

**Proof 2.** Next, we want to prove that this new noise corresponds to our intuitive definition of a warped noise. For this, let $\mathbf{x} \in E$ and consider $\Delta \mathbf{x} \in \mathcal{E}$ a tiny subset around $\mathbf{x}$. By setting $A = \mathcal{T}(\Delta \mathbf{x})$ in the definition of $\widetilde{W}$, we have:

$$
\widetilde{W}(\mathcal{T}(\Delta \mathbf{x})) = \int_{\mathbf{y} \in \mathcal{T}(\Delta \mathbf{x})} \frac{1}{|\nabla \mathcal{T}(\mathcal{T}^{-1}(\mathbf{y}))|^{\frac{1}{2}}} W(\mathcal{T}^{-1}(\mathbf{y})) d\mathbf{y}
\tag{28}
$$

By the same change of variable as before we get:

$$
\widetilde{W}(\mathcal{T}(\Delta \mathbf{x})) = \int_{\mathbf{y} \in \Delta \mathbf{x}} |\nabla \mathcal{T}(\mathbf{y})|^{\frac{1}{2}} W(\mathbf{y}) d\mathbf{y}.
\tag{29}
$$

By linearization, we assume that the determinant of the warping field's Jacobian is constant over $\Delta \mathbf{x}$, replacing it by its mean yields

$$
\begin{aligned}
\widetilde{W}(\mathcal{T}(\Delta \mathbf{x})) &\simeq \int_{\mathbf{y} \in \Delta \mathbf{x}} \left( \frac{1}{\nu(\Delta \mathbf{x})} \int_{\mathbf{u} \in \Delta \mathbf{x}} |\nabla \mathcal{T}(\mathbf{u})| \, d\mathbf{u} \right)^{\frac{1}{2}} W(\mathbf{y}) d\mathbf{y} \\
&= \left( \frac{1}{\nu(\Delta \mathbf{x})} \int_{\mathbf{u} \in \Delta \mathbf{x}} |\nabla \mathcal{T}(\mathbf{u})| \, d\mathbf{u} \right)^{\frac{1}{2}} \left( \int_{\mathbf{y} \in \Delta \mathbf{x}} W(\mathbf{y}) d\mathbf{y} \right) \\
&= \left( \frac{1}{\nu(\Delta \mathbf{x})} \int_{\mathbf{v} \in \mathcal{T}(\Delta \mathbf{x})} d\mathbf{v} \right)^{\frac{1}{2}} \left( \int_{\mathbf{y} \in \Delta \mathbf{x}} dB(\mathbf{y}) \right) \\
&= \left( \frac{\nu(\mathcal{T}(\Delta \mathbf{x}))}{\nu(\Delta \mathbf{x})} \right)^{\frac{1}{2}} W(\Delta \mathbf{x}),
\end{aligned}
\tag{30}
$$

which concludes the proof. $\qquad\square$

### C.3 FROM CONTINUOUS TO DISCRETE FORMULATION

In practice, it is infeasible to warp a continuous noise field with the noise transport equation. We will demonstrate that Equation (4) becomes Equation (5) once discretized. Our goal is to compute the pixel value $G(\mathbf{p})$ in the warped noise for the given pixel coordinates $\mathbf{p} = (i, j)$. Using our $\int$-noise interpretation, $G(\mathbf{p})$ corresponds to the integral of the underlying warped white noise, i.e. $G(\mathbf{p}) = \mathcal{T}(W)(A)$, where $A = [i-1, i] \times [j-1, j]$ is the subset of the domain $E$ covered by the pixel at position $\mathbf{p}$. Thus, substituting $G(\mathbf{p})$ in Equation (4):

$$
G(\mathbf{p}) = \int_{\mathbf{x} \in A} \frac{1}{|\nabla \mathcal{T}(\mathcal{T}^{-1}(\mathbf{x}))|^{\frac{1}{2}}} W(\mathcal{T}^{-1}(\mathbf{x})) \, d\mathbf{x}
\tag{31}
$$

Assuming that the warping field has a locally constant Jacobian over the entire pixel area around $\mathbf{p}$, we can apply the approximation from Equation (25) with $\Delta \mathbf{x} = A$:

$$G(\mathbf{p}) \simeq \left(\frac{\nu(A)}{\nu(\mathcal{T}^{-1}(A))}\right)^{\frac{1}{2}} W(\mathcal{T}^{-1}(A))$$

$$= \frac{1}{\nu(\mathcal{T}^{-1}(A))^{\frac{1}{2}}} W(\mathcal{T}^{-1}(A)) \qquad \text{since } \nu(A) = 1 \text{ by definition of } A. \tag{32}$$

The second approximation comes from rasterization. We approximate the warped pixel shape by its rasterized version at level $k$. This is formally written as

$$\mathcal{T}^{-1}(A) \simeq \bigcup_{A_i^k \in \Omega_\mathbf{p}} A_i^k, \tag{33}$$

where $\Omega_\mathbf{p} \subseteq \mathbb{A}^k$ contains all subpixels at level $k$ that are covered by the warped pixel polygon $\mathcal{T}^{-1}(A)$ after rasterization. Substituting this into $G(\mathbf{p})$ and using linearity of $\nu$ and $W$, we get

$$G(\mathbf{p}) \simeq \frac{1}{\nu\left(\bigcup_{\Omega_\mathbf{p}} A_i^k\right)^{\frac{1}{2}}} W\left(\bigcup_{\Omega_\mathbf{p}} A_i^k\right)$$

$$= \frac{1}{\left(\sum_{A_i^k \in \Omega_\mathbf{p}} \nu(A_i^k)\right)^{\frac{1}{2}}} \sum_{A_i^k \in \Omega_\mathbf{p}} W(A_i^k)$$

$$= \sqrt{\frac{N_k}{|\Omega_p|}} \sum_{A_i^k \in \Omega_\mathbf{p}} W(A_i^k) \qquad \text{since } \nu(A_i^k) = 1/N_k, \tag{34}$$

$$= \frac{1}{\sqrt{|\Omega_p|}} \sum_{A_i^k \in \Omega_\mathbf{p}} W_k(A_i^k) \qquad \text{since } W_k(A_i^k) = \sqrt{N_k} W(A_i^k),$$

which concludes the derivation. $\qquad\square$

It is easy to verify that the resulting $G(\mathbf{p})$ remains a standard normal Gaussian variable, as it is the "Gaussian average" of multiple standard normal variables. Furthermore, the independence between different pixels $G(\mathbf{p})$ and $G(\mathbf{q})$ is ensured with a proper implementation of rasterization, i.e. if rasterization of adjacent triangles are disjoint.

Lastly, note that all the approximations we made to go from the continuous setting to the discrete one only affect the accuracy of the deformation field (locally constant Jacobian, approximate warped region), but never the actual *values* of the noise samples. This is key to the distribution-preserving properties of our method.

### C.4 IMPLEMENTATION

We include the pseudo-code of our $\int$-noise warping for a single pixel in Algorithm (2), and provide more details on the algorithm itself regarding specific aspects such as warping and the treatment of disocclusions.

**Warping.** To warp the triangulation points $V$ for the pixel contour, we follow the common practice in physics simulations. As the deformation map $\mathcal{T}$ is only defined at pixel centers, we use bicubic interpolation to obtain the deformation map values at the sub-pixel triangulation points.

**Handling disocclusions.** When employing optical flow maps as the deformation mapping $\mathcal{T}$ between the anchor and current frame, some pixels can end up with an undefined value. There are two possible causes to this issue. The first one is that the pixel polygon was stretched in a way that it ended up covering zero sub-pixels after rasterization. This can happen when the deformation is large, or the sub-pixels are not fine enough (i.e. $k$ is too low). The second one is that two pixel polygons are rasterized one on top of the other. As we only keep the values of the last one, no sub-pixel

---

**Algorithm 2** Distribution-preserving noise warping (for a single pixel)

---

**Input:** $G$: discrete noise at anchor frame (in size $D \times D$)
    $A$: pixel area in current frame
    $\mathcal{T}$: deformation mapping between the anchor and current frame
    $k$: noise upsampling factor
    $s$: polygon subdivision steps
**Output:** pixel value $x$ in current frame

$G_\infty \leftarrow \text{UPSAMPLE}_\infty(G, k)$
$(V, F) \leftarrow \text{TRIANGULATE\_AREA}(A, s)$
$V \leftarrow \text{WARP}_\infty(V, \mathcal{T})$
$\Omega \leftarrow \text{RASTERIZE}((V, F), G_\infty)$
$x \leftarrow \sum_{(i,j) \in \Omega} G_\infty(i, j) / \sqrt{\text{SIZE}(\Omega)}$

---

is associated with the first polygon. This typically happens if there are disocclusions and the flow map is not diffeomorphic.

To solve this in practice, we use a two-stage process. First, we warp the noise from the *initial* frame and fill in the current frame's pixels. At this stage, some pixels might be undefined. At the second stage, we perform another warping, this time from the *previous* frame (up-sampled on the fly), and use it to fill the remaining missing values. If there are still missing values left, we replace them by randomly sampled noise. This allows the best temporal consistency for most of the pixels.

### C.5 1-D Toy Example: Proof

In Section 2.2, we gave an intuition on why our $\int$-noise is able to preserve the distribution using a 1-D toy example. Here we aim at proving Equation 6 in a similar way of proving the conditional white noise sampling presented in Appendix B.2.

First, let us consider a continuously defined noise signal $W$ over the 1-D segment $E = [0, 2]$. We further define two independent random variables $\mathbf{x}_0 = W([0, 1]), \mathbf{x}_1 = W([1, 2])$ representing two neighboring "pixels". A random variable $\mathbf{z}$ interpolating linearly between $\mathbf{x}_0$ and $\mathbf{x}_1$ in space can be defined as $\mathbf{z} = W([\alpha, 1 + \alpha])$, with $\alpha \in [0, 1]$. Then, all three variables have a variance of 1 and $\mathbf{z}$ is correlated with $\mathbf{x}_0$ and $\mathbf{x}_1$ through

$$\begin{aligned}
\text{Cov}(\mathbf{z}, \mathbf{x}_0) &= \nu([\alpha, 1 + \alpha] \cap [0, 1]) = \nu([\alpha, 1]) = 1 - \alpha, \\
\text{Cov}(\mathbf{z}, \mathbf{x}_1) &= \nu([\alpha, 1 + \alpha] \cap [1, 2]) = \nu([1, 1 + \alpha]) = \alpha.
\end{aligned} \tag{35}$$

Thus, merging the variables into a vector $\mathbf{Y} = (\mathbf{z}, \mathbf{x}_0, \mathbf{x}_1)^\top$ yields

$$\mathbf{Y} \sim (\boldsymbol{\mu}, \boldsymbol{\Sigma}), \quad \text{with } \boldsymbol{\mu} = \mathbf{0}, \ \boldsymbol{\Sigma} = \begin{bmatrix} 1 & 1 - \alpha & \alpha \\ 1 - \alpha & 1 & 0 \\ \alpha & 0 & 1 \end{bmatrix}. \tag{36}$$

Applying the conditional probability rule for a Gaussian vector (as in Appendix B.2) gives:

$$\mathbf{z} | \mathbf{x}_0 = x_0, \mathbf{x}_1 = x_1 \sim \mathcal{N}\left(\bar{\mu}, \bar{\sigma}^2\right), \quad \text{where} \begin{cases} \bar{\mu} = (1 - \alpha, \alpha) \cdot (x_0, x_1)^\top, \\ \bar{\sigma}^2 = 1 - (1 - \alpha, \alpha) \cdot (1 - \alpha, \alpha)^\top. \end{cases} \tag{37}$$

Simplifying the expressions yields

$$\bar{\mu} = (1 - \alpha) x_0 + \alpha x_1, \quad \text{and} \quad \sigma^2 = 1 - (\alpha^2 + (1 - \alpha)^2), \tag{38}$$

which concludes the proof.

### C.6 Comparisons against other warping approaches

**Interpolation baselines.** Our method enables accurate Gaussian noise warping, even when the noise undergoes substantial deformations. Given a two dimensional warping field extracted from

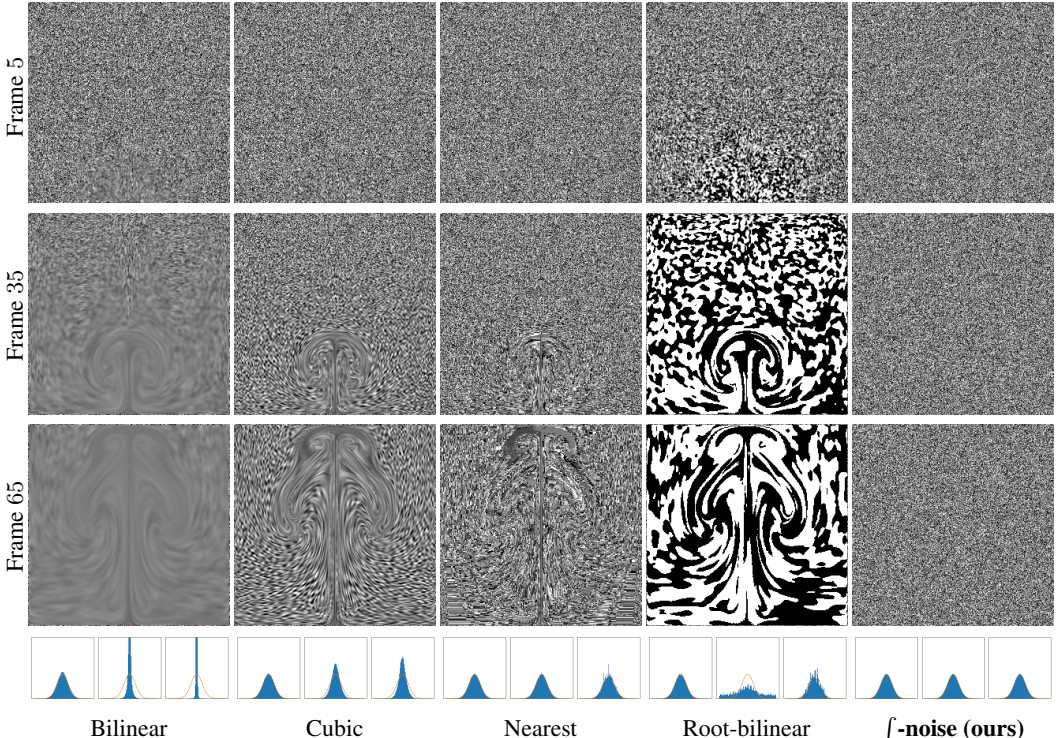

Figure 7: Warping a noise sample with different schemes. Rows indicate different frames, while columns represent the method used for transport. The noise sequence is warped incrementally, transporting one frame at the time with a given velocity field. Warping through bilinear or bicubic interpolation quickly blurs the noise image. Nearest neighbor produces better results, but introduces artifacts due to incorrect spatial correlations. Root-bilinear interpolation has the proper distribution at the start, but produces incorrect results once the pixels are no longer independent. Our proposed $\int$-noise is able to properly maintain temporal correlations introduced by warping while also preserving the original signal distribution.

the fluid simulation dataset (Figure 5), Figure 7 shows a comparative study between various warping methods. Three classical types of interpolation were tested: bilinear, bicubic and nearest neighbor. While bilinear and bicubic interpolations compromise high-frequency details, nearest neighbor creates artifacts in the form of duplicate samples, since it is oblivious to space deformations introduced by the warping field. We also experimented with a simple modification of the bilinear interpolation that is able to preserve the variance of the pixels by replacing the weighting coefficients with their square roots. Though being efficient, this approach still introduces spatial correlations that also eventually lead to artifacts. Our noise method outperforms all existing warping methods by transporting the noise perfectly while keeping its Gaussian properties.

**Noise prior baselines.** We also compared the proposed $\int$-noise warping against previous noise priors such as the residual-based noise sampling of Control-A-Video (Chen et al., 2023), and the mixed and progressive noise model from PYoCo (Ge et al., 2023). The residual-based sampling keeps the noise signal fixed for image pixels that remain unchanged in a video, while the noise is resampled in locations in which temporal variations of RGB values go over a predefined threshold. PYoCo introduces two sampling schemes: mixed and progressive. Mixed noise sampling linearly combines a frame-dependent noise sample and a noise shared across all frames. Progressive noise sampling generates a sequence of noise samples in an autoregressive fashion: the noise at the current frame is generated by perturbing the noise from the previous frame. These approaches are not able to preserve the exact spatial correlations of the original sequence, since they are not based on warping. We show a comparison between different warping methods for the task of super-resolution

| Method | Wall Time | CPU Time |
|---|---|---|
| Random | 0.01 | 0.01 |
| Fixed | 0.01 | 0.01 |
| PYoCo (mixed) | 0.01 | 0.01 |
| PYoCo (prog.) | 0.01 | 0.01 |
| Control-A-Video | 6.08 | 95.46 |
| Bilinear | 5.26 | 76.76 |
| Bicubic | 6.00 | 87.73 |
| Nearest | 5.17 | 75.73 |
| Root-bilinear | 7.66 | 103.78 |
| DDIM Inv. (20 Steps) | 853.42 | 2226.6 |
| DDIM Inv. (50 Steps) | 2125.5 | 3608.3 |
| $\int$-noise (ours, $k = 3, s = 4$) | 629.01 | 2274.9 |

Table 3: **Runtime Comparisons of Different Noise Schemes.** The measurements are in milliseconds per frame at resolutions $256 \times 256$.

| | $s = 1$ | $s = 2$ | $s = 3$ | $s = 4$ |
|---|---|---|---|---|
| $k = 0$ | 21.6 | 21.7 | 23.0 | 26.2 |
| $k = 1$ | 23.6 | 23.5 | 23.8 | 25.1 |
| $k = 2$ | 30.5 | 29.3 | 29.6 | 30.8 |
| $k = 3$ | 58.8 | 55.4 | 55.0 | 53.7 |
| $k = 4$ | 143.8 | 137.5 | 132.4 | 128.2 |

| | $s = 1$ | $s = 2$ | $s = 3$ | $s = 4$ |
|---|---|---|---|---|
| $k = 0$ | 10.5 | 10.6 | 11.3 | 12.9 |
| $k = 1$ | 10.5 | 10.6 | 10.8 | 11.3 |
| $k = 2$ | 11.2 | 11.3 | 11.5 | 11.9 |
| $k = 3$ | 15.5 | 15.3 | 15.5 | 15.6 |
| $k = 4$ | 29.1 | 29.2 | 28.6 | 28.7 |

Table 4: **CPU time (top) and wall time (bottom) of $\int$-noise computation for different $k$ and $s$ parameters.** The measurements are for a video sequence of 24 frames at resolution $256 \times 256$, in seconds.

in Figures 14 and 15. We refer to our supplementary website for additional comparisons between different warping techniques.

### C.7 RUNTIME COMPARISONS

We estimate both wall time and CPU time for all methods on a video sequence at resolution $256 \times 256$ in Table 3. By evaluating the run time for different sub-sequence lengths, we can linearly regress the average per frame computation time of each method. This is evaluated on a GeForce RTX 3090 GPU and an Intel i9-12900K CPU.

Simple noise sampling methods like PYoCo Ge et al. (2023), fixed or random noise can be executed efficiently on GPU. Methods that rely on information from the input sequence in a simple way such as Control-A-Video Chen et al. (2023) or the interpolation methods are 3 orders of magnitude slower than the simple noise sampling methods. Finally, our warping method requires $\approx 1$ second per frame to warp the noise. While being considerably less efficient than aforementioned methods, our approach is comparable to DDIM inversion.

Additionally, our method trades off the noise warp accuracy with efficiency through two parameters: $k$, the noise upsampling factor, and $s$, the polygon subdivision steps. We report the CPU and wall clock time in seconds for noise warping in a video sequence of 24 frames at resolution $256 \times 256$ for different $k$ and $s$ in Table 4.

### D ADDITIONAL RESULTS

In this section, we provide more results and evaluations for the applications shown in Section 3.

### D.1 PHOTOREALISTIC APPEARANCE TRANSFER WITH SDEDIT

Figure 8 shows different variations of photorealistic appearance transfer from the same synthetic video sequence (top row). Contrary to DDIM inversion, our temporally-correlated warping scheme can be applied to arbitrary noise samples, allowing us to generate different edits from the same input sequence. In Figure 10, we compare our $\int$-noise warping against a fixed noise sample on a close-up scene. When using a fixed noise sample, the original fold in the corner of the bed stays at the same position in the image even after the bed rotates; our method, on the other hand, successfully transports the fold, following the bed movement. Other comparisons for the bedroom example are shown in Figure 9 and 12. In Table 5, we quantitatively evaluate our method on different bedroom scenes. Our noise warping method not only outperforms standard Gaussian noise priors in terms of temporal

| Method | w/o cross-frame attention | | | | w/ cross-frame attention | | | |
|---|---|---|---|---|---|---|---|---|
| | scene 1 | scene 2 | scene 3 | mean | scene 1 | scene 2 | scene 3 | mean |
| Random | 12.85 | 11.82 | 11.21 | 11.96 | 10.02 | 10.23 | 9.74 | 10.00 |
| Fixed | 4.75 | 4.19 | 5.44 | 4.79 | 4.40 | 3.90 | 4.74 | 4.35 |
| PYoCo (mixed) | 9.25 | 8.28 | 8.44 | 8.65 | 7.67 | 6.79 | 7.31 | 7.26 |
| PYoCo (prog.) | 6.27 | 5.52 | 6.41 | 6.06 | 5.18 | 4.26 | 5.25 | 4.90 |
| Control-A-Video | 5.13 | 5.48 | 6.08 | 5.56 | 4.65 | 4.71 | 5.91 | 5.09 |
| Bilinear | **3.12** | 3.93 | 8.33 | 5.13 | 2.70 | 1.89 | 4.86 | 3.15 |
| Bicubic | 3.24 | 5.97 | 15.10 | 8.10 | **2.67** | 3.05 | 9.12 | 4.95 |
| Nearest | 13.77 | 28.78 | 46.59 | 29.71 | 5.12 | 13.69 | 26.48 | 15.10 |
| $\int$-noise (ours) | 3.47 | **2.09** | **3.68** | **3.08** | 2.97 | **1.70** | **2.83** | **2.50** |

Table 5: **Temporal coherency metrics for the task of appearance transfer with SDEdit.** Each value is averaged over a batch of 4 different variations. The metric is the masked MSE between the current frame and the warped previous frame averaged over the entire sequence (units are in $\times 10^{-3}$).

| Method | w/o cross-frame attention | | w/ cross-frame attention | |
|---|---|---|---|---|
| | FID | Precision | FID | Precision |
| Random | 61.14 | 0.659 | 74.75 | 0.719 |
| Fixed | 79.32 | 0.622 | 93.56 | 0.644 |
| PYoCo (mixed) | 65.07 | 0.656 | 81.60 | 0.674 |
| PYoCo (prog.) | 66.43 | 0.635 | 84.63 | 0.667 |
| Control-A-Video | 75.21 | 0.618 | 90.82 | 0.649 |
| Bilinear | 185.28 | 0.101 | 143.24 | 0.201 |
| Bicubic | 168.86 | 0.118 | 149.85 | 0.212 |
| Nearest | 192.13 | 0.177 | 154.73 | 0.344 |
| $\int$-noise (ours) | 73.83 | 0.655 | 92.63 | 0.661 |

Table 6: **Sample quality metrics for the task of appearance transfer with SDEdit.** We measure perceptual quality of the generated bedroom frames. The FID (Heusel et al., 2017) metric measures sample quality and diversity at the same time. The precision metric (Kynkäänniemi et al., 2019) measures sample quality through the likelihood of our generated samples to belong to the LSUN Bedroom dataset. We use the implementation from Dhariwal & Nichol (2021).

coherency, it also manages to beat standard interpolation methods. The reason for this is visualized in Figure 11: standard interpolation methods are unable to account for space deformations, which leads to artifacts in the denoised results.

We evaluate the visual quality of the generated frames in Table 6 using FID (Heusel et al., 2017) and Improved Precision (Kynkäänniemi et al., 2019). FID (Heusel et al., 2017) is computed between the LSUN Bedroom dataset and the set containing all images from a given video example. It is important to note that FID simultaneously measures sample quality and diversity, while we are only interested in quality. Since we are computing FID over a sequence of frames from the same video, the diversity is naturally very low, which explains the overall high values we have in Table 6 (typical FID values are less than 50). Additionally, less temporally coherent methods like *Random Noise* will likely have more variations between frames, which improves diversity. This explains why random noise performs the best in terms of FID, and our method typically performs slightly worse. All in all, the main take-away from the FID comparison is that our method performs in the same range as other Gaussian noise priors (random, fixed, PYoCo, Control-A-Video) and much better than the other interpolation methods. Improved Precision (Kynkäänniemi et al., 2019) is a metric of sample quality only, as it measures the likelihood of the generated samples to belong to the ground truth dataset manifold. As the table shows, our method performs on par with other Gaussian noise priors in terms of visual quality.

We additionally compared our noise prior with DDIM inversion in Figure 13. As DDIM cannot be directly used with DDPM-based methods like SDEdit, we devise two reasonable ways of comparing

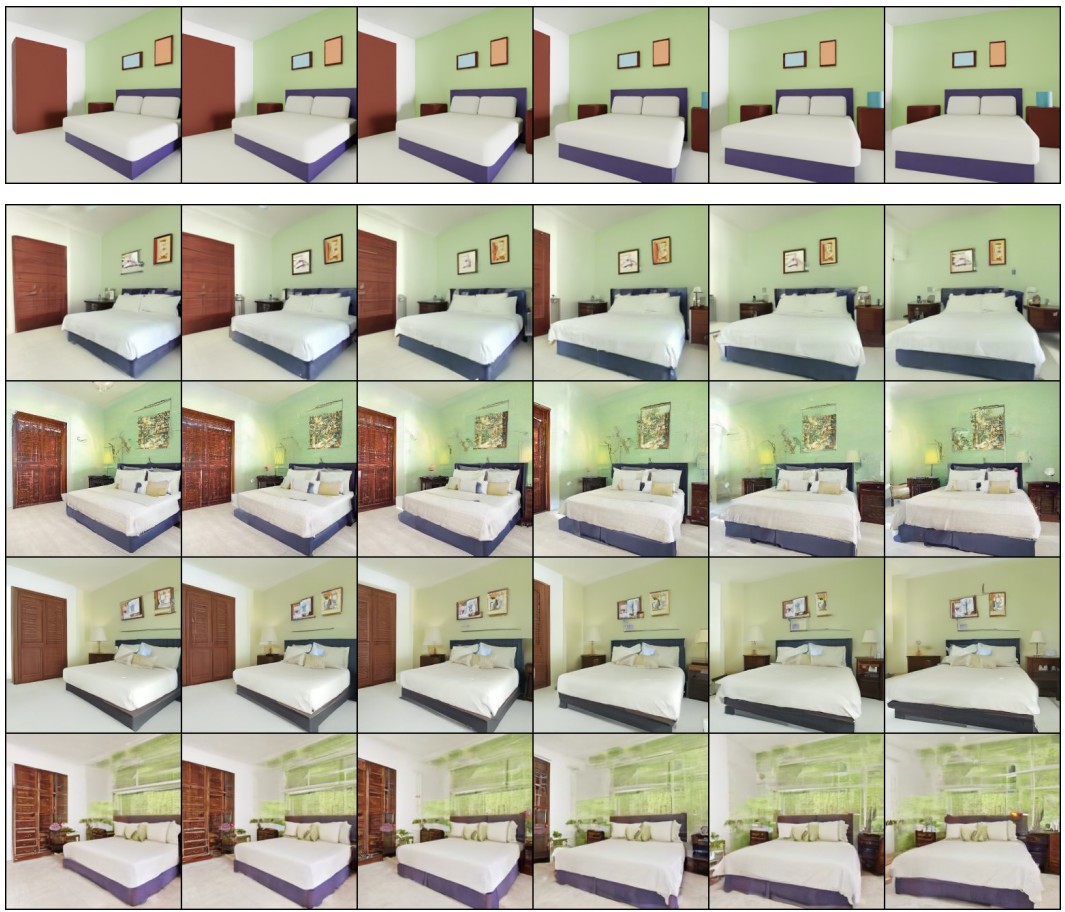

Figure 8: In combination with cross-frame attention, noise warping helps create temporally-coherent realistic rendering of a room from a synthetic scene. Top row: input synthetic scene, other rows: results obtained using $\int$-noise with different initial noise.

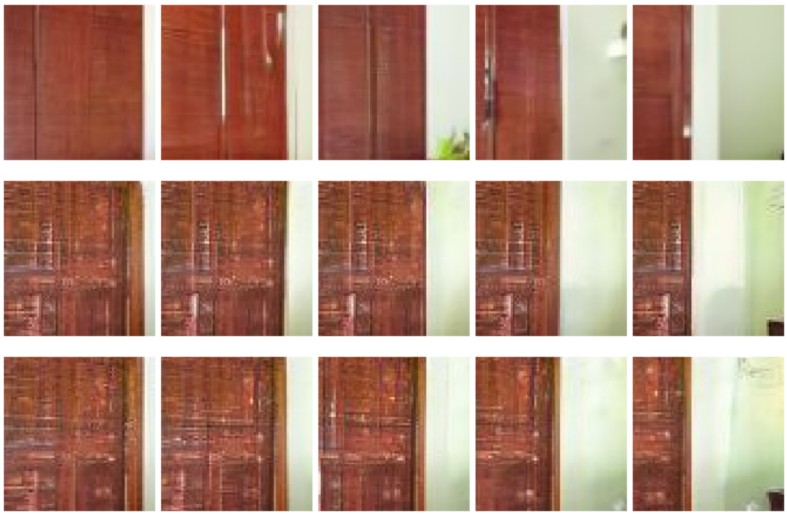

Figure 9: While random noise creates different details at each frame (top), fixing the noise leads to the exact same details in image space (middle). Noise warping successfully translates the details with the object (bottom).

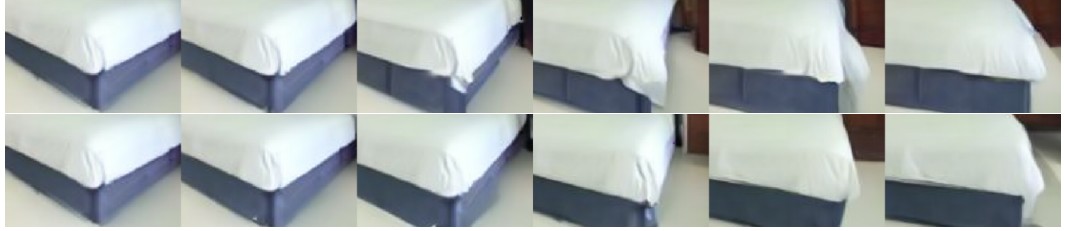

Figure 10: Using the same noise at every frame (top) leads to sticking artifacts, as shown by the bedsheet fold staying in the same spot of the image. Noise warping mitigates that (bottom).

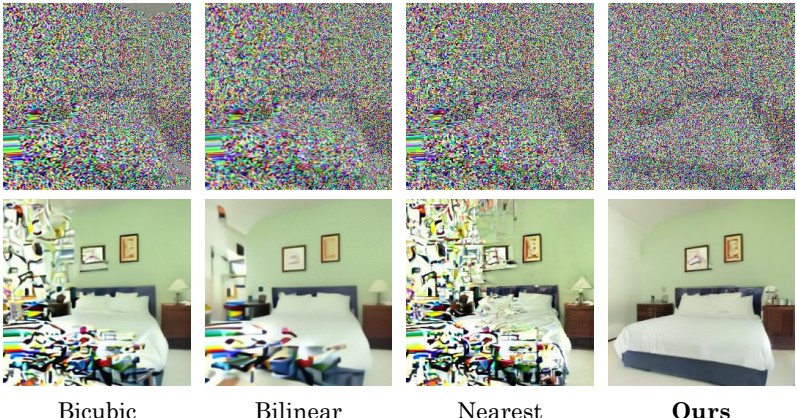

| Bicubic | Bilinear | Nearest | **Ours** |

Figure 11: Our method has quantitatively better temporal coherency than traditional warping methods, since large motions cause unaccounted noise stretching for classical interpolation (top row), which leads to visible artifacts in the denoised image (bottom row).

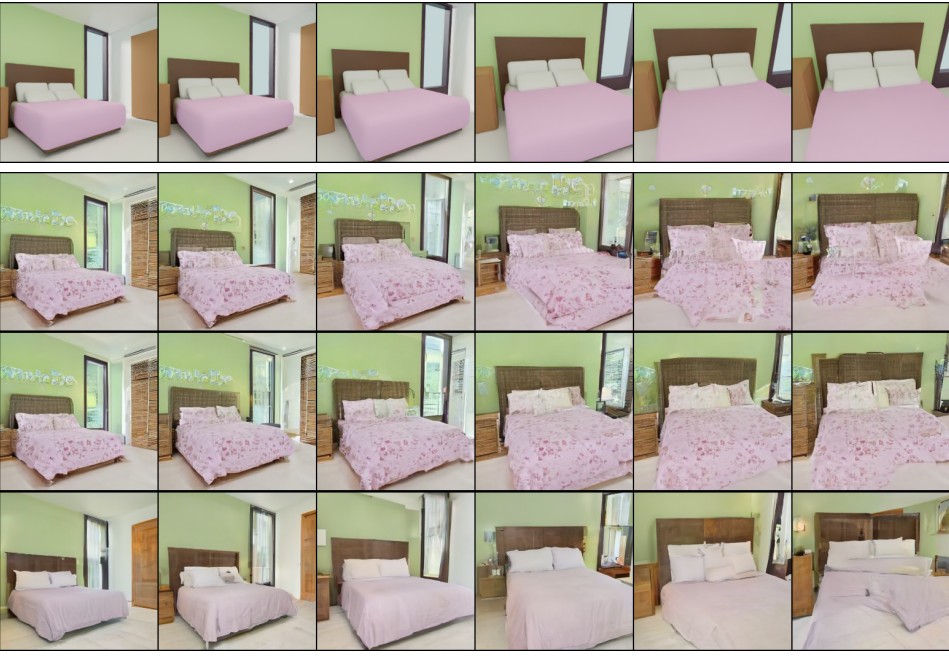

Figure 12: Comparing $\int$-noise (top), fixed noise (center), and random noise (bottom) for a different bedroom example.

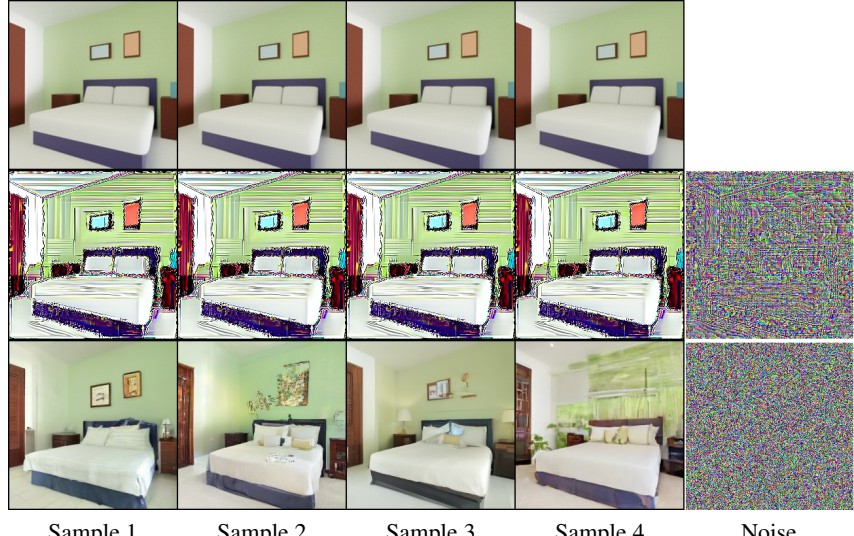

Sample 1    Sample 2    Sample 3    Sample 4    Noise

Figure 13: Comparing DDIM inversion to intermediate step (top), DDIM inversion as initial noise (middle), and our $\int$-noise prior (bottom).

with DDIM noise in photorealistic appearance transfer. In the first experiment (Figure 13, top row), we apply DDIM inversion to intermediate step (60% of total steps) and then denoise it using forward DDIM. As expected, since neither the prompt or settings are changed, this approach mostly reconstructs the original synthetic video without adding any realistic appearance details. In the second experiment, DDIM inversion is employed to obtain the initial noise (Figure 13, middle row), we then add it to input frames and denoise using forward DDIM. Because the input video is far from the data distribution of the model (trained on realistic images of bedrooms), the DDIM-inverted noise is far from Gaussian. This makes it a poor candidate as a noise prior. As DDIM remains primarily an inversion method, the spatial and temporal information of the image are entangled inside the noise. On the contrary, our noise prior only contains temporal information, so the model can generate realistic details on top of the synthetic scene without being constrained to reconstruct the input sequence.

## D.2    VIDEO RESTORATION WITH I$^2$SB

We evaluate our noise prior on two video restoration tasks based on I$^2$SB, namely super-resolution and JPEG compression restoration. Table 7 and 8 respectively evaluate the temporal coherency and the image quality for different noise priors. Our $\int$-noise prior performs worse in JPEG restoration

| Warp MSE $_{(\times 10^{-3})}$ ↓ | $4\times$ super-resolution | | | | JPEG restoration | | | |
|---|---|---|---|---|---|---|---|---|
| | bear | blackswan | car turn | mean | cows | goat | train | mean |
| Random | 3.98 | 9.57 | 13.17 | 8.91 | 6.71 | 11.36 | 6.78 | 8.28 |
| Fixed | 2.18 ● | 8.69 ● | 13.03 | 7.97 ● | 5.15 | 11.21 | 6.25 | 7.54 |
| PYoCo (mixed) | 3.09 | 9.28 | 13.08 | 8.48 | 5.84 | 11.12 | 6.35 | 7.77 |
| PYoCo (prog.) | 2.32 | 9.08 | 12.90 ● | 8.10 | 5.33 | 11.05 | 6.06 | 7.48 |
| Control-A-Video | 2.24 ● | 8.88 ● | 12.82 ● | 7.98 ● | 5.73 | 11.16 | 6.31 | 7.73 |
| Bilinear | 9.96 | 25.52 | 28.94 | 21.47 | **2.85** ● | **8.44** ● | **4.49** ● | **5.26** ● |
| Bicubic | 4.60 | 15.79 | 18.66 | 13.02 | 3.34 ● | 8.99 ● | 4.90 ● | 5.74 ● |
| Nearest | 2.95 | 17.38 | 22.56 | 14.30 | 5.03 | 11.14 | 7.55 | 7.91 |
| $\int$-**noise (ours)** | **1.19** ● | **6.76** ● | **11.51** ● | **6.49** ● | 4.04 ● | 8.88 ● | 5.01 ● | 5.98 ● |

Table 7: **Temporal coherency analysis for I$^2$SB video tasks.** We show temporal coherency results for I$^2$SB video tasks evaluated on the DAVIS dataset. The metric is the masked MSE between the current frame and the warped previous frame averaged over the entire sequence (units are in $\times 10^{-3}$).

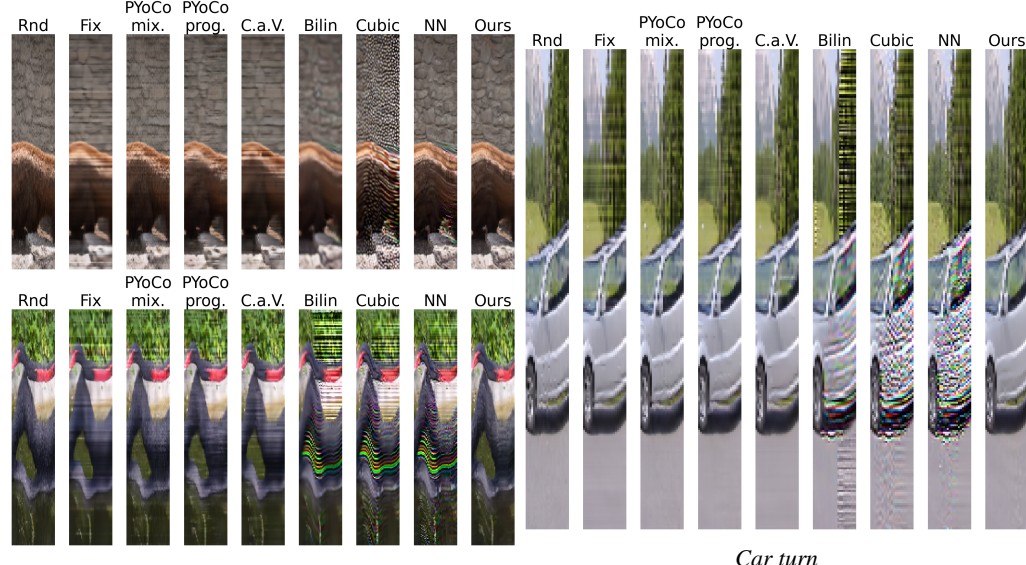

*Bear & Blackswan*

*Car turn*

Figure 14: **Visualizing $x$-$t$ slices of the I$^2$SB video super-resolution task.** We visualize the different methods from the quantitative evaluation (in the same order). Our noise prior (right) does not exhibits stripes that are manifestations of sticking artifacts. It also seems less blurry than standard interpolation methods, and less noisy than Random Noise.

relative to super-resolution. This might be due to the optical flow being harder to estimate from the corrupted sequence in the first case.

In Figure 14 and Figure 15, we visualize temporal slices of the denoised videos by concatenating a single column from each frame. In this visualization, sticking artifacts manifests as horizontal stripes, making it easy to evaluate temporal coherency qualities. The various sequences show that warping the noise consistently yields temporally-smoother and spatially-sharper results.

| LPIPS ↓ | 4× super-resolution | | | | JPEG restoration | | | |
| --- | --- | --- | --- | --- | --- | --- | --- | --- |
| | bear | blackswan | car turn | mean | cows | goat | train | mean |
| Random | 0.242 | 0.179 ○ | 0.154 | 0.192 | 0.157 | **0.201** ● | **0.131** ● | **0.163** ● |
| Fixed | **0.224** ● | 0.180 | **0.134** ● | **0.179** ● | **0.151** ● | **0.201** ● | 0.136 | **0.163** ● |
| PYoCo (mixed) | 0.242 | 0.179 ○ | 0.149 ○ | 0.190 ○ | 0.156 | 0.207 | 0.136 | 0.166 |
| PYoCo (prog.) | 0.243 | **0.172** ● | 0.156 | 0.190 ○ | 0.153 ○ | 0.205 | 0.132 ○ | **0.163** ● |
| Control-A-Video | 0.240 ● | 0.183 | 0.154 | 0.192 | 0.156 | 0.205 | 0.132 ○ | 0.164 |
| Bilinear | 0.668 | 0.589 | 0.515 | 0.590 | 0.447 | 0.513 | 0.332 | 0.431 |
| Bicubic | 0.627 | 0.500 | 0.343 | 0.490 | 0.358 | 0.452 | 0.305 | 0.372 |
| Nearest | 0.297 | 0.414 | 0.275 | 0.329 | 0.167 | 0.253 | 0.221 | 0.213 |
| ∫-noise (ours) | 0.235 ○ | 0.202 | 0.152 ● | 0.196 | 0.154 ● | 0.202 ● | 0.138 | 0.165 |

Table 8: **Image perceptual quality analysis for I$^2$SB video tasks.** We measure perceptual quality of the generated results for I$^2$SB video tasks evaluated on sequences from the DAVIS dataset. The use LPIPS (↓) with VGG network to assess the visual quality of the images. Standard interpolation methods leads to blurred results, which affect the perceptual quality. Our noise prior performs in the same range as standard Gaussian noise.

## D.3 Pose-to-person video generation with PIDM

When employing the Person Image Diffusion Model (PIDM) (Bhunia et al., 2022) for generating videos relative to an input posed person sequence, the pose sequence is not enough to estimate accurate motion vectors. So we run a few steps of diffusion with PIDM to get a blurry early prediction of the person, from which we then extract the full body pose with DensePose (Güler et al., 2018). The

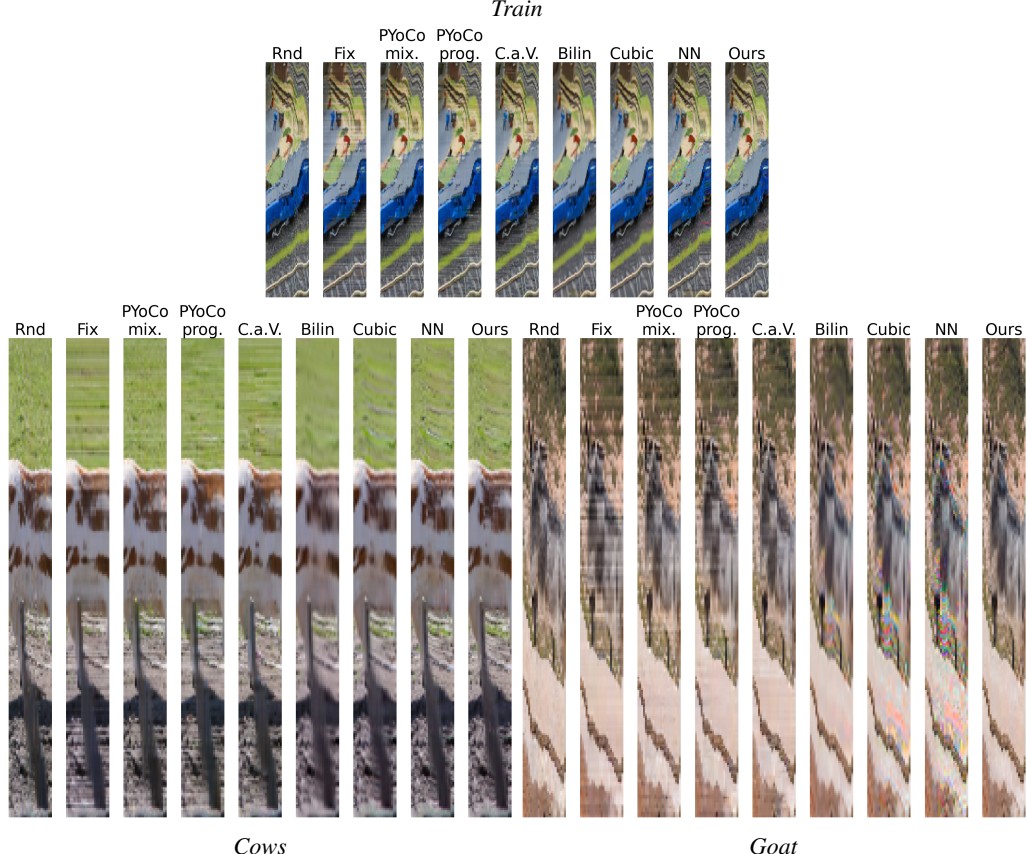

Figure 15: **Visualizing $x$-$t$ slices of the I$^2$SB video JPEG restoration task.**

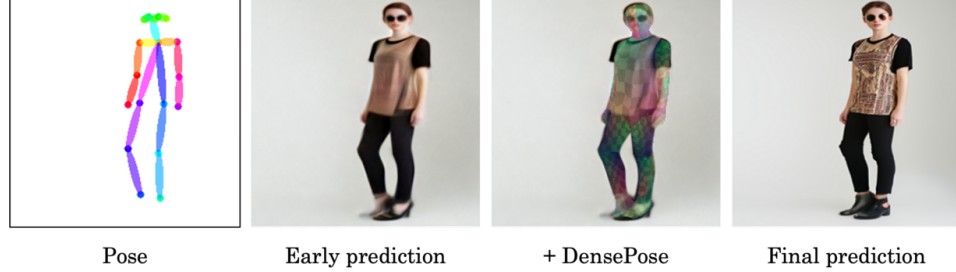

Figure 16: Pipeline for motion estimation using PIDM.

DensePose result is overlaid on top of the early prediction as the final prediction before computing the optical flow (Teed & Deng, 2020). The different stages of the pipeline are depicted in Figure 16.

We get test examples from the DeepFashion dataset (Liu et al., 2016). Table 9 gives a more detailed quantitative evaluation regarding temporal coherency. In particular, we notice that in the examples evaluated in Section 3, the pose sequences generally have little motion (as is often the case when fashion models pose). This could explain why fixing the seed has a better temporal coherency. We alternatively test on pose sequences where we simulate the camera moving, causing the poses to shift horizontally. The results are reported in the table under "High-variance pose". Similarly, fixing the random seed gives fairly good temporal coherency. This is in contradiction with the visual results, which clearly shows sticking artifacts. A possible reason comes from the fact that we compute the temporal coherency on the entire image, and most of the background has zero motion, which may bias the mean warp error to favor the fixed seed experiments.

| Warp MSE $(\times 10^{-3})\downarrow$ | Quasi-static pose | | | | High-variation pose | | | |
|---|---|---|---|---|---|---|---|---|
| | input 1 | input 2 | input 3 | mean | input 1 | input 2 | input 3 | mean |
| Random | 26.40 | 46.50 | 23.72 | 32.21 | 27.01 | 50.13 | 25.53 | 34.22 |
| Fixed | 2.40 ● | 3.29 ● | 3.14 ● | 2.94 ● | **1.76** ● | **2.22** ● | 2.78 ● | 2.26 ● |
| PYoCo (mixed) | 15.81 | 22.46 | 12.43 | 16.90 | 18.78 | 28.73 | 15.25 | 20.92 |
| PYoCo (prog.) | 9.73 | 15.11 | 8.28 | 11.04 | 11.92 | 15.06 | 8.91 | 11.97 |
| Control-A-Video | 6.34 | 13.90 | 8.47 | 9.57 | 6.41 | 7.00 | 5.94 | 6.45 |
| Bilinear | **1.77** ● | **3.00** ● | **2.87** ● | **2.55** ● | 1.83 ● | 2.30 ● | **2.28** ● | **2.13** ● |
| Bicubic | 3.15 | 4.07 ● | 4.10 | 3.77 | 2.15 ● | 2.41 ● | 2.66 ● | 2.40 ● |
| Nearest | 11.69 | 17.83 | 12.73 | 14.08 | 7.21 | 13.21 | 7.20 | 9.21 |
| $\int$-noise (ours) | 2.54 ● | 4.58 | 3.52 ● | 3.55 ● | 2.42 | 3.03 | 3.29 | 2.92 |

Table 9: **Temporal coherency analysis for the pose-to-person video with PIDM task.** We show temporal coherency results for pose-to-person video with PIDM using input images from the Deep-Fashion dataset.

## D.4 FLUID SUPER-RESOLUTION

Our last application is on the super-resolution of 2-D fluid simulations. We train an unconditional diffusion model on a large dataset of fluid simulations generated with the fluid solver from Tang et al. (2021) without special treatment on the noise. We then compare different noise priors, including the one from Chen et al. (2023). The results are hard to assess frame by frame. Thus, we encourage the reader to take a look at the supplementary videos.

## E NOISE WARPING IN LATENT DIFFUSION MODELS

### E.1 UNDERSTANDING THE LIMITATIONS

Noise warping does not improve temporal coherency in latent diffusion models as much as one would expect. We pinpoint this observation to three main reasons. First, temporally coherent images do not necessarily translate into temporally coherent autoencoder latent vectors. Thus, temporally consistent noise priors in the VAE latent space might be suboptimal. Second, the noise used in latent diffusion models does not contribute directly to the fine details of the results in image space, which is where warped noise priors excel. Lastly, the autoencoder almost always introduces a non-negligible reconstruction error that affects the temporal coherency of the results, independent of the latent diffusion process.

From a more theoretical perspective, the temporally-correlated infinite-resolution noise warping we propose is only possible because of the structural self-similarity of Gaussian noise, which allows us to interpret a discrete Gaussian noise sample as an aggregated view of more Gaussian samples at a smaller scale. The warping could thus be operated at the limit, in the continuous setting. This core assumption does not hold in latent diffusion models, because temporal coherency is no longer targeted in the latent space where the noise resides, but rather in the image space mapped by the decoder. Unfortunately, Gaussian noise decoded by the VAE no longer possesses the self-similarity property. This aspect can be visualized in the decoded noise shown by Figure 17a (a) (iii).

Nonetheless, the assumption that moving the noise helps with temporal coherency still holds in latent models to some extent, as some of our experiments below can show. One of our early observations was that the VAE decoder is translationally equivariant in a discrete way, i.e. translating the latent vector by an integer number of pixels leads to an almost perfectly shifted image when decoded. For an autoencoder with a compression factor $f = 8$ (e.g. Stable Diffusion), this means that a 1-pixel shift in the latent space would produce a 8-pixel shift in the final image.

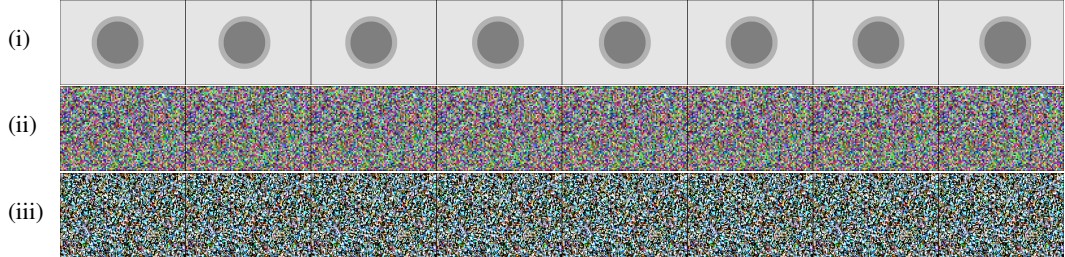

(a) Inputs to the latent diffusion experiment: (i) input frames, (ii) warped latent noise, (iii) decoded by VAE.

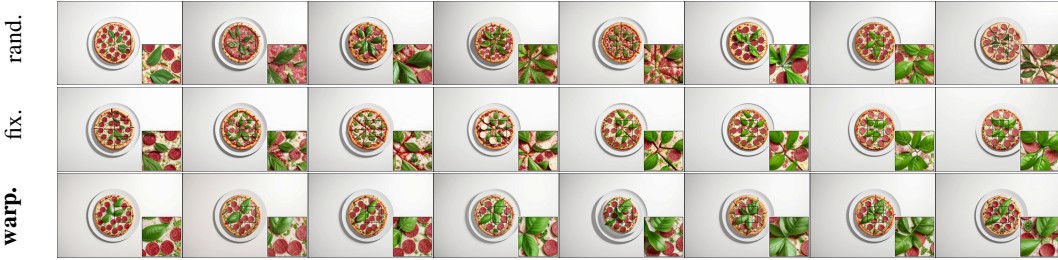

(b) Video-to-video translation baseline: frame by frame w/ ControlNet and a prompt.

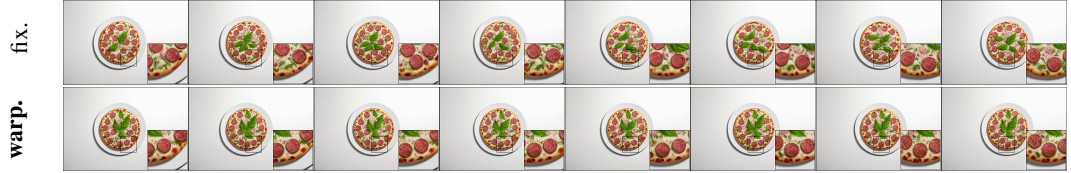

(c) Video-to-video translation: (b) + w/ cross-frame attention.

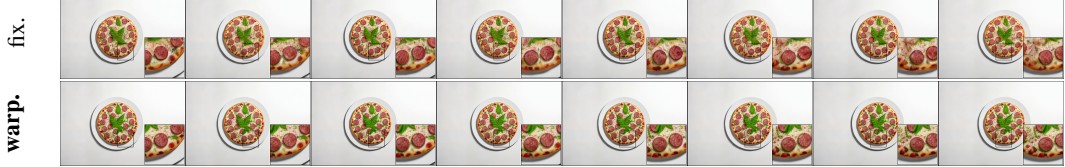

(d) Video-to-video translation: (c) + w/ feature injection.

Figure 17: **Noise warping in latent diffusion models.** As the latent diffusion model gets more constrained, through ControlNet and a prompt (b), + cross-frame attention (c), + feature map injection (d), results improve in temporal coherency. In the final example (d), translating the noise along with the content gives a pixel-perfect translating pizza, while keeping the noise fixed still induces sticking artifacts on the high-frequency details. Please refer to the supplemental videos for a better visualization of the differences between these examples.

### E.2 Experiment setup

With this in mind, we generated a simple video consisting of two concentric gray discs translating by 8 pixels to the right every frame, in front of a white background (see Figure 17a (a) (i)). We add noise to each frame, and denoise it using Stable Diffusion with the following prompt:

*"Commercial photography, square salami pizza, italian food, white lighting, top view, studio light, 8k octane rendering, high resolution photography, insanely detailed, fine details, on white isolated plain, 8k, commercial photography, stock photo, professional color grading"*

We manually simulate the warped Gaussian noise used in latent space by moving a disc of noise in front of a noisy background (Figure 17a (a) (ii)). The input frames are used as "scribbles" with ControlNet and the prompt provides the necessary information to generate the results. This sequence, as many of our other results, is better visualized in the included supplementary videos.

### E.3 Results and analysis

We perform three sets of experiments, each time with increasing constraints on the model. All the results reveal empirically that our $\int$-noise prior is better, provided it is warped properly.

**Baseline:** using the input frames with ControlNet and the given prompt, we generate results frame by frame (Figure 17b). While all the noise schemes produce temporally inconsistent results, flickering is especially visible when using random noise at each frame (top row). For fixed random seed (middle row), the insets show an overall structure is fixed in image space, even though the pizza is translating. This can be seen be paying attention to the regions covered by the basil leaves, which remains rougly the same. Finally, warping the noise with the input video improves the movement of the synthesized details (bottom row). In particular, the basil leaves progressively invades the inset, indicating proper translation.

**Cross-frame attention:** a common trick used to boost temporal-coherency in diffusion-based video-to-video translation tasks is to replace all self-attention layers by cross-attention with respect to an anchor keyframe (Hertz et al., 2022; Parmar et al., 2023). Figure 17c shows the results of additionally adding cross-frame attention with respect to the first frame for every frame. This effectively creates more constraints for the model, leading to similar content between frames. Nonetheless, fixing the noise (top row) still creates sticking artifacts, mainly on the position of the salami slices in image space. Another visible artifact can be seen in the position of the darker spots of the pizza crust, which stays in place. On the contrary, moving the noise with the input generates much more natural motion of the ingredients on the pizza (bottom row).

**Feature map injection:** another tool for improving temporal coherency involves injecting feature maps (Tumanyan et al., 2022) from one image to the other. This is usually done in addition to employing cross-frame attention. It can be applied between frames, as long as feature maps are also warped. By injecting the feature maps from the first frame for all layers 4-11, our noise warping method produces pixel-perfect translation of the generated pizza (Figure 17d bottom row). Keeping the noise fixed, however, induces sticking artifacts of fine details (top row).

**Analysis**. As shown by the experiments, the impact of the noise scheme is negatively correlated with the amount of constraints given to the model. As most of the low-frequency structures gets dictated by the cross-frame attention and the feature maps, the noise becomes solely responsible for the fine high-frequency details. In any case, noise warping is a necessary treatment tor provide temporal coherency to diffusion models when they are employed on a frame by frame basis.

