# OpenReview forum: "How I Warped Your Noise: a Temporally-Correlated Noise Prior for Diffusion Models"
_ICLR.cc/2024/Conference — ICLR 2024 oral_

### Official Review · Reviewer_tVGv · 2023-10-16

**Soundness:** 4 excellent
**Presentation:** 4 excellent
**Contribution:** 4 excellent
**Rating:** 6
**Confidence:** 4

**Summary:**

This paper studied the influence of initialized noise when adopting diffusion based image model to video model.

A new initialization method called integral noise is propose, which is composed of a conditional upsampling, rasterization, and aggregation stage. It can maximizing the correlation between the warped and the original sample and maintain the independence of pixels in each sample.

**Strengths:**

The method is well illustrated in Figure 2 and is easy to understand.
A deep and comprehensive theoretical analysis is provided in the method and supplementary material.
The method is evaluated on several tasks, including rerendering (SDEdit), pose-to-person, and video restoration.
Good video illustration in supplementary material.

**Weaknesses:**

I guess most experiments in this paper are conducted in zero-shot way which applies the image processing model and method to video processing problems. Looking forward to seeing whether this helps when training a video model, like the PYoCo model
Considering video restoration (**Zeroscope Text-to-Video**) and pose-to-human (dreampose) already have good performance, I wonder the gap between image model+ integrate noise and these video models.
As stated in Appendix E, integrat noise does not work well in latent diffusion model. I wonder if this is applicable to other pixel-level diffusion models like the open-sourced deepfloyd from stabilityAI, whose capacity is comparable or better than stable diffusion.

**Questions:**

I am happy to see the comparison with the video processing model and the integration with DeepFloyd

---

> ### Author Response · Authors · 2023-11-21
> **Reply to Reviewer tVGv**
>
> We thank the reviewer for the constructive feedback. We will address the remaining questions below.
>
> ---
>
> **Zero-shot and training**
>
> >I guess most experiments in this paper are conducted in zero-shot way which applies the image processing model and method to video processing problems. Looking forward to seeing whether this helps when training a video model, like the PYoCo model.
>
> This is correct, we propose our noise warping method primarily as a novel tool to improve temporal coherency of video editing tasks in a zero-shot setting. This is part of an effort to leverage pretrained image models for video tasks without requiring additional resource-intensive training of models on video datasets. However, as the reviewer mentioned, employing this temporally-correlated noise prior for training or fine-tuning video diffusion models is definitely an exciting direction for future work, and we will be happy to see adoption of our $\int$-noise in these contexts.
>
> ---
>
> **Comparison with video models**
>
> >Considering video restoration (Zeroscope Text-to-Video) and pose-to-human (dreampose) already have good performance, I wonder the gap between image model+ integrate noise and these video models.
>
> In our work, we raised awareness on the importance of noise priors for temporal consistency and focused on solving the mathematical problem of distribution-preserving Gaussian noise warping. We then validated our proposed solution on several zero-shot video editing tasks with diffusion models, making it an effective tool for improving temporal coherency in these applications.
>
> While our proposed method is compatible with other zero-shot techniques like cross-frame attention (which we implemented in some of our results), it is challenging to directly compare our method alone with video models for several reasons.
>
> First, as many previous work on zero-shot video editing with image diffusion models have shown, it is often necessary to combine different techniques together to achieve good temporal coherency. For instance, Pix2Video [1] uses a combination of DDIM inversion, loss guidance and depth conditioning in their pipeline. A fair comparison against existing video models would have to be task-specific (super-resolution, stylization, video generation etc.), as different works adopt different pipelines. Our approach is a general method to enforce noise priors for diffusion models, and could be easily incorporated into different pipelines.
>
> Secondly, a comparison with these video models requires at least similar inputs to be fair. In the example of pose-to-person, Dreampose [2] is trained to take a very informative DensePose [3] parameterization as input, while we based our method on PIDM which only takes an Openpose poses [4]. Note that we did additionally use DensePose in our application results, but only for motion estimation (see Appendix D.3).
>
> All in all, we believe combining our $\int$-noise with other techniques like depth conditioning can further improve the quality of the results.
>
> On another note, we have added more visual comparisons of our methods with other noise priors such as Control-A-Video and DDIM inversion, which we believe can be helpful. Please refer to the new project webpage for additional results: https://warpyournoise.github.io/
>
> ---
>
> **Integration with DeepFloyd IF**
>
> >I wonder if this is applicable to other pixel-level diffusion models like the open-sourced deepfloyd from stabilityAI, whose capacity is comparable or better than stable diffusion.
>
> Our noise warping method is also applicable to other non-latent diffusion models like DeepFloyd from StabilityAI. We added some examples of using our noise warping with the open-source model for the task of video super-resolution and stylization to support our claim. The results can be visualized on our project webpage: https://warpyournoise.github.io/#deepfloyd We thank the reviewer for pointing this out, as this broadens the generalization abilities of our method beyond the category specific models we showcased in our initial manuscript.
>
> ---
>
> **Conclusion / References**
>
> We hope this clarified some of the interrogations. Please feel free to reach out for any further questions/comments regarding the submission and our answers above.
>
> References:
>
> [1] Ceylan, Duygu, Chun-Hao P. Huang, and Niloy J. Mitra. "Pix2video: Video editing using image diffusion." Proceedings of the IEEE/CVF International Conference on Computer Vision. 2023.
>
> [2] Karras, Johanna, et al. "Dreampose: Fashion image-to-video synthesis via stable diffusion." arXiv preprint arXiv:2304.06025 (2023).
>
> [3] Güler, Rıza Alp, Natalia Neverova, and Iasonas Kokkinos. "Densepose: Dense human pose estimation in the wild." Proceedings of the IEEE conference on computer vision and pattern recognition. 2018.
>
> [4] Cao, Zhe, et al. "Realtime multi-person 2d pose estimation using part affinity fields." Proceedings of the IEEE conference on computer vision and pattern recognition. 2017.

---

> > ### Comment · Reviewer_tVGv · 2023-11-21
> > **Reply to Author**
> >
> > Thanks so much for your great work! The shared results are intereseting.

---

### Official Review · Reviewer_H5Nc · 2023-10-21

**Soundness:** 3 good
**Presentation:** 3 good
**Contribution:** 3 good
**Rating:** 8
**Confidence:** 5

**Summary:**

This paper proposes a method for warping Gaussian noise while preserving its Gaussian properties.
To achieve the goal, the authors consider a mathematical model of the Brownian sheet (i.e. the distributional derivative of the two-dimensional Brownian motion) and develop a noise transport equation for that model.
In applications, the following approximation is applied. First, the input noise map is conditionally up-scaled so that it remains Gaussian. Second, the up-scaled map is warped according to the equation derived for the Brownian sheet.
Finally, the warped noise is down-scaled to the original size.

The method is tested on a number of vision problems. According to the provided evaluations, typically it produces better temporal coherence than the baseline methods.

**Strengths:**

From my point of view, this paper tackles a pretty important problem for the community, since the application of image translation models to videos is a popular direction which still suffers from flickering and other artifacts. It seems to me that the Brownian sheet model and conditional Gaussian upscaling both are a very good fit for that problem as they take into account the intrinsically hierarchical nature of image resolution. The video results in the supplementary demonstrate that the amount of flickering and, vice versa, texture sticking in videos is reduced.

**Weaknesses:**

1. From the theoretical point of view, the presented approach relies on the warping field being a diffeomorphism, while in practice to the best of my knowledge optical flows estimated with off-the-shelf methods are rarely invertible mappings.
1. As the authors themselves admit (Appendix E.3), "the impact of the noise scheme is negatively correlated with the amount of constraints given to the model", and, in particular, the method has a limited impact on latent diffusion models. However, it is obviously still useful for cascaded models.
1. As mentioned in the paper (Sec. 6) the method is computationally less efficient than other techniques.
1. According to Tab. 1, while the proposed method typically increases temporal coherence, at the same time it worsens frame-wise image plausibility metrics such as LPIPS or FID.
1. None of the video samples in the supplementary except for fluid dynamics provides the results of noise handling with the method proposed by Control-A-Video model. Please note that this model more close to the presented method in spirit since it also takes input video into account while PYoCo does not.

**Questions:**

1. As indicated in Tab. 1, bilinear interpolation often results in quite decent quality (excl. Video SR task). Haven't the authors considered a small modification of bilinear interpolation which is able to preserve the variance of pixels, namely, replacing the commonly used weighting coefficients with their square roots? Probably, this simple modification can even further improve its performance. Also note that Eq. 5 is also essentially "Gaussian averaging" of subpixels, i.e. averaging using variance-preserving weighting coefficients of $1/\sqrt{k}$ instead of $1/k$.
1. As mentioned above, the presented method is less computationally efficient than the baselines. It would be nice to provide a quantitative evaluation of that (in)efficiency.
1. From the manuscript, it is a bit unclear how the 9 triangulation points are warped in Fig. 1a. How exactly is the estimated optical flow $\mathcal{T}^{-1}$ which typically has the same resolution as the image itself, applied to non-integer pixel coordinates? In other words, how is the operation $\textrm{WARP}_{\infty}$ from Algorithm 2 implemented?
1. I suggest adding an explicit proof of why $\textbf{Z}$ is a Gaussian vector (Appendix B.2). Probably, the shortest proof is the one considering the linear combinations $\kappa^T \textbf{Z}$ for all possible $\kappa$, which are obviously Gaussian random variables.
1. I recommend adding more comments on how Eq. 24 for the continuous case turned into Eq. 5 for the discrete case. Why is it valid to still rely on the first-order approximation?

---

> ### Author Response · Authors · 2023-11-16
> **Reply to Reviewer H5Nc (1/2)**
>
> We thank the reviewer for the constructive feedback. We will address the weaknesses mentioned below.
>
> ---
>
> **Treatment of non-diffeomorphic warping fields**
>
> >From the theoretical point of view, the presented approach relies on the warping field being a diffeomorphism, while in practice to the best of my knowledge optical flows estimated with off-the-shelf methods are rarely invertible mappings.
>
> It is true that the theoretical derivation relies on the warping field being a diffeomorphism, which is generally not the case for optical flow maps. In particular, disocclusions can reveal empty spaces that aren’t mapped to any part of the initial frame.
>
> We solve this in practice by replacing missing values with newly sampled noise. On very long sequences, we also resample the high-resolution noise periodically, which effectively updates the anchor frame. We will add a more detailed explanation regarding how we handle warping in practical settings in the Appendix.
>
> ---
>
> **Image quality measure in quantitative evaluation**
>
> > According to Tab. 1, while the proposed method typically increases temporal coherence, at the same time it worsens frame-wise image plausibility metrics such as LPIPS or FID.
>
> For the appearance transfer example, we used FID as a measure of image quality, since there is no ground truth in this case. FID is computed between the LSUN Bedroom dataset and a dataset we create from frames of the video results we have. Unfortunately, FID simultaneously measures sample quality and diversity, while we are only interested in quality. Since we are computing it over a sequence of frames from the same video, the diversity is naturally very low, which explains the overall high values we have in Tab. 1 (typical FID values are <50). Additionally, less temporally coherent methods like ‘Random Noise’ will likely have more variations between frames, which improves diversity. This explains why random noise performs the best wrt. FID, and our method typically performs a bit worse. All in all, the main take-away from the FID comparison is that our method performs in the same range as other Gaussian noise priors (random, fixed, PYoCo, Control-A-Video) and much better than the other interpolation methods. This, in complement with the warp error metric, shows that we are able to take the “best of both worlds”.
>
> Regarding the LPIPS metric for video super-resolution and restoration, we do observe a slight decrease in image quality according to the metrics, which we cannot fully explain. Similarly, the main take-away is that we are performing in the same range as the other noise priors methods, while having much better temporal coherency.
>
> ---
>
> **More comparisons with Control-A-Video**
>
> > None of the video samples in the supplementary except for fluid dynamics provides the results of noise handling with the method proposed by Control-A-Video model. Please note that this model more close to the presented method in spirit since it also takes input video into account while PYoCo does not.
>
> The residual noise from Control-A-Video behaves like fixed noise in regions of uniform color, and like random noise around moving edges between objects. Our experiments have shown that this generally leads to the “worst of both worlds'', as both flickering and texture sticking happens. We have updated the results with more comparisons with the residual noise prior. They can be found on our project webpage: https://warpyournoise.github.io/
>
> ---

---

> > ### Author Response · Authors · 2023-11-16
> > **Reply to Reviewer H5Nc (2/2)**
> >
> > In the following, we will continue with addressing the questions raised by the reviewer.
> >
> > ---
> >
> > **Root-bilinear interpolation**
> >
> > > As indicated in Tab. 1, bilinear interpolation often results in quite decent quality (excl. Video SR task). Haven't the authors considered a small modification of bilinear interpolation which is able to preserve the variance of pixels, namely, replacing the commonly used weighting coefficients with their square roots? Probably, this simple modification can even further improve its performance.
> >
> > We have indeed considered the exact same modification in our early experiments! It seemed like a very simple idea for solving our problem, but the experiments revealed some issues with this method:
> > * First, this averaging will only preserve the variance if it averages samples that are independent. This means we cannot simply warp from the previous frame as in standard bilinear interpolation. Rather, we have to go back to the first frame as in our proposed method.
> > * Second, even by doing so, such a variance-preserving bilinear interpolation would still create correlations between neighboring pixels in the warped noise. This translates to some loss of high frequency details. Therefore, while it preserves the variance, it does not preserve the independence of the pixels.
> > * Lastly, we tried using these in diffusion models. And because the noise has less high frequency but still has unit variance, the diffusion model is unable to denoise it properly. This leads to even worse artifacts than simple bilinear interpolation.
> >
> > We included a visualization of the noise advected with the root-bilinear approach, which can be found at https://warpyournoise.github.io/#noise-comp. Additionally, we reported measurements of efficiency for this approach in our [answer to Reviewer 2 (3JNo)](https://openreview.net/forum?id=pzElnMrgSD&noteId=nVUOv3C1dX).
> >
> > ---
> >
> > **Computational efficiency**
> >
> > > As mentioned above, the presented method is less computationally efficient than the baselines. It would be nice to provide a quantitative evaluation of that (in)efficiency.
> >
> > Our method is indeed computationally less efficient than most other noise prior methods. This is a known trade-off that we decided to make, as our intent was to solve Gaussian noise warping as accurately as we can to see how far that could bring us. Interestingly, it is still comparable to DDIM inversion in runtime. Additionally, our method has two parameters $k$ and $s$ which control the resolution of the upsampled noise and the discretization level of the pixel polygons respectively. These parameters provide users with the possibility to trade off accuracy in the temporal correlations with efficiency.
> >
> > Please refer to our [answer to Reviewer 2 (3JNo)](https://openreview.net/forum?id=pzElnMrgSD&noteId=nVUOv3C1dX) for further details regarding the method’s efficiency.
> >
> > ---
> >
> > **Warping of triangulation points**
> >
> > > From the manuscript, it is a bit unclear how the 9 triangulation points are warped in Fig. 1a. How exactly is the estimated optical flow which typically has the same resolution as the image itself, applied to non-integer pixel coordinates? In other words, how is the operation from Algorithm 2 implemented?
> >
> > Following standard practices in physics simulation of fluids, we warp the triangulation points by upsampling the flow map with bicubic interpolation. We will include this in the updated manuscript.
> >
> > ---
> >
> > **Editing suggestions**
> >
> > > I suggest adding an explicit proof of why is a Gaussian vector (Appendix B.2). Probably, the shortest proof is the one considering the linear combinations for all possible, which are obviously Gaussian random variables.
> >
> > We will integrate this into the updated manuscript.
> >
> >
> >
> > > I recommend adding more comments on how Eq. 24 for the continuous case turned into Eq. 5 for the discrete case. Why is it valid to still rely on the first-order approximation?
> >
> > It is indeed an important aspect of our contribution, we will clarify this in a new section in the Appendix. Intuitively the discretization does not affect the theoretical guarantees because we are only approximating the true warped pixel polygon with a "discretized" one. But the set of all warped polygons still creates a partition of the domain, so all the guarantees on independence and variance preservation still hold. In a sense, our re-interpretation of noise as a continuous field allows us to transfer the error from value space (e.g. bilinear interpolation) to the geometrical representation of the domain (approximation on the shape of the polygon).
> >
> > ---
> >
> > **Conclusion**
> >
> > We hope this clarified some of the interrogations. We will update the paper to integrate the feedback. Please feel free to reach out for any further questions/comments regarding the submission and our answers above.

---

> > ### Comment · Reviewer_H5Nc · 2023-11-17
> >
> > Thanks a lot for the prompt response!
> >
> > > Unfortunately, FID simultaneously measures sample quality and diversity, while we are only interested in quality.
> >
> > This is true, so I would like to bring to your attention the well-adopted metrics which measure the "quality" of the images. Probably, the best known is the Improved Precision [1]. Density [2] is similar although less adopted by the community.
> >
> > [1] Kynkäänniemi et al. Improved Precision and Recall Metric for Assessing Generative Models. In NeurIPS, 2019.
> >
> > [2] Naeem et al. Reliable Fidelity and Diversity Metrics for Generative Models. In ICML, 2020.

---

### Official Review · Reviewer_3JNo · 2023-10-29

**Soundness:** 3 good
**Presentation:** 3 good
**Contribution:** 3 good
**Rating:** 8
**Confidence:** 2

**Summary:**

This work attempts to mitigate the lack of temporal correlation seen in diffusion based video generation models. The proposed method attempts to generate new noise samples that preserve the correlations induced by motion vectors. To this end, first, this work reinterprets individual noise samples used in diffusion models as a continuously integrated noise field called integral noise. With the help of the derived noise transport equation, a transport algorithm is developed to generate noise with temporal correlation between samples while preserving the desired properties of noise samples. Results are demonstrated to indicate the potential of the proposed method for tasks such as video restoration and editing, surrogate rendering, and conditional video generation.

**Strengths:**

1. The problem tackled in this work has significant practical impact.
2. The paper is written very well.
3. Motivation is clear, ideas are well formulated with theory, and the results are impressive.

**Weaknesses:**

1. The proposed method is computationally inefficient as compared to prior arts. Quantitative comparisons on this aspect would have been more helpful for future research works.

**Questions:**

1. Figure 5 illustration is hard to follow, it’s not clear how to judge the performance based on the images shown. It’s mentioned that, the Random Noise creates incoherent details while Fixed Noise suffers from sticking artefacts. Our R -noise moves the fluid in a smoother way. However, I find this explanation hard to map to the results shown. Maybe the authors can highlight based on the image contents the reasons for each of these conclusions.

---

> ### Author Response · Authors · 2023-11-16
> **Reply to Reviewer 3JNo**
>
> We thank the reviewer for the constructive feedback. We will address the remaining questions below.
>
> ---
>
> **Computational Efficiency**
>
> > The proposed method is computationally inefficient as compared to prior arts. Quantitative comparisons on this aspect would have been more helpful for future research works.
>
> We estimate both wall time and CPU time for all methods on a video sequence from the surrogate rendering application (resolution at 256x256). By evaluating the run time for different sub-sequence lengths, we can linearly regress the average per frame computation time of each method. This is evaluated on a GeForce RTX 3090 GPU and an Intel i9-12900K CPU. The table below summarizes the run time of our method against previous approaches. We will add this table to the updated manuscript appendix.
>
> | Method | Wall Time (in ms/frame) | CPU Time (in ms/frame) |
> | :-- | :--: | :--: |
> | Random | 0.01| 0.01|
> | Fixed | 0.01  | 0.01|
> | PYoCo (mixed) [1] | 0.01| 0.01 |
> | PYoCo (progressive) [1]  | 0.01| 0.01|
> | Control-A-Video [2]| 6.08   | 95.46|
> | Bilinear  | 5.26| 76.76|
> | Bicubic | 6.00| 87.73|
> | Nearest| 5.17| 75.73 |
> | Root-bilinear| 7.66 |103.78|
> | DDIM inversion (20 steps) [3] | 853.42 | 2226.6|
> | DDIM inversion (50 steps) [3] | 2125.5  | 3608.3 |
> | **$\int$﻿-noise (ours, k=3, s=4)**| **981.76**| **2953.6**|
>
> Our method is indeed less efficient. This is a trade-off that we decided to make, as our intent was to solve Gaussian noise warping as accurately as possible, to understand how far that could bring us. It is worth noting that we have explored several simpler, less computationally heavy ways to warp the noise by making different simplifying assumptions (such as the root-bilinear interpolation methods shown in the table, a modification also suggested by Reviewer 3 (H5Nc) in Question 1). However, none of those led to satisfying solutions.
>
> An analysis of the table shows that simple noise sampling methods like PYoCo, fixed or random noise can be executed efficiently on GPU. Methods that rely on information from the input sequence in a simple way such as Control-A-Video or interpolation methods are almost 3 orders of magnitude slower than the simple noise sampling methods. Finally, our warping method requires ~1s per frame to warp the noise. While being considerably less efficient than aforementioned methods, the proposed approach is comparable to DDIM inversion. With a standard setting of 50 DDIM steps, DDIM inversion takes longer than our method to produce a noise map, while being similar to ours in run time if we reduce the number of steps to 20. Please refer to our answer to Reviewer 1 (13WQ) for more qualitative comparisons with DDIM inversion.
>
> | k \ s | 1| 2| 3| 4|
> | -- | -- | -- | -- | -- |
> | **0**| 10.0 | 10.3| 10.7| 12.1 |
> | **1**| 10.1 | 10.5| 10.9| 12.2 |
> | **2**| 10.3 | 10.5| 10.9| 12.3 |
> | **3**| 12.2 | 12.5| 13.0| 14.2 |
> | **4**| 17.9 | 17.9 | 18.4  | 19.5 |
>
> Additionally, our method has two parameters $k$ and $s$ which control the resolution of the upsampled noise and the discretization level of the pixel polygons respectively. These parameters provide users with the possibility to trade off accuracy in the temporal correlations against efficiency. We refer to the table above to demonstrate how the parameters impact the performance. The measurements represent the wall clock time in seconds that our method requires to compute noise for a video sequence of 24 frames at resolution 256x256. Additionally, we refer to the Section 4 of the paper on how these parameters affect the quality of the results.
>
> Lastly, our implementation was not optimized for efficiency. We believe it can still be drastically improved, which would make the computational cost of using our noise scheduler negligible.
>
> ---
>
> **Better visual comparison for fluid super-resolution**
>
> >Figure 5 illustration is hard to follow, it’s not clear how to judge the performance based on the images shown.
>
> Unfortunately it is hard to show video results properly on paper. We will update the visualization in the submission to reflect the feedback. Additionally, the video results of this figure can be found on our project webpage at https://warpyournoise.github.io/#fluid-sr, where the visual comparison is much clearer.
>
> ---
>
> **Conclusion / References**
>
> We hope this clarified some of the interrogations. We will update the paper to integrate the feedback. Please feel free to reach out for any further questions/comments regarding the submission and our answers above.
>
> References:
>
> [1] Ge, Songwei, et al. "Preserve your own correlation: A noise prior for video diffusion models." Proceedings of the IEEE/CVF International Conference on Computer Vision. 2023.
>
> [2] Chen, Weifeng, et al. "Control-A-Video: Controllable Text-to-Video Generation with Diffusion Models." arXiv preprint arXiv:2305.13840 (2023).
>
> [3] Song, Jiaming, Chenlin Meng, and Stefano Ermon. "Denoising diffusion implicit models." arXiv preprint arXiv:2010.02502 (2020).

---

> > ### Comment · Reviewer_3JNo · 2023-11-18
> > **Comment on Author Response**
> >
> > Many thanks for the detailed response and clarifications. Authors have addressed all my concerns satisfactorily.

---

### Official Review · Reviewer_13WQ · 2023-11-01

**Soundness:** 4 excellent
**Presentation:** 3 good
**Contribution:** 3 good
**Rating:** 8
**Confidence:** 4

**Summary:**

This paper introduces the novel problem of Gaussian noise warping. Therein, the authors raised the interesting question of "how to properly warp the Gaussian noise such that the warped noise map still has the Gaussian distribution preserved?". To this end, the authors discussed why applying conventional warping operation on the noise map is not suitable, and they proposed a mathematically grounded solution to the problem with their ∫-noise formulation. The authors motivated the practical value of this problem with the applications of lifting image diffusion models to perform temporally consistent video editing. Experiment results with different video editing tasks demonstrate the effectiveness of the proposed method.

**Strengths:**

The research question “how to properly warp a Gaussian noise map” introduced by this paper is interesting scientifically.

The problem is well motivated with clear explanations on why the problem is not trivial, i.e. why the conventional warping operations are not suitable. The proposed solution is technically sound and well-grounded mathematically.

The practical aspects of the problem are also well-motivated, with interesting video editing applications.

**Weaknesses:**

The major weakness from the practical point of view is the implicit assumption that temporally correlated noise maps can induce temporally consistent video editing results, which is often not true. This limitation, however, has been acknowledged and explained by the authors in the paper.

**Questions:**

Other than the comments above, there are a couple of questions I’m curious about:
+ I’m wondering how sensitive the method is with respect to the underlying estimated flow map? In other words, how do the errors in the estimated optical flows affect the results?
+ One principled way to obtain a sequence of correlated noise maps is to perform inverse DDIM on the reference video frames (assuming such video is available, which is true in most of the use cases demonstrated in this paper). The authors did mention that technique in the paper, but did not elaborate further and did not show visual results or comparison. I’m wondering how will the noise maps obtained that way compare to the ones obtained from the ∫-noise in terms of the final video quality?

---

> ### Author Response · Authors · 2023-11-15
> **Reply to Reviewer 13WQ**
>
> We thank the reviewer for the constructive feedback. We will address the remaining questions below.
>
> ---
>
> **Implicit assumption on temporally-coherent noises**
>
> > The major weakness from the practical point of view is the implicit assumption that temporally correlated noise maps can induce temporally consistent video editing results, which is often not true.
>
> While there is no theoretical guarantee that diffusion models produce temporally-consistent video editing results with a given temporally-correlated noise map, we observe that this is often the case in practice. Take the translating pizza example on Appendix E: if we add a noise that undergoes the same translation as the input video, the generated images are more temporally coherent. For better visualization and understanding of these results, we refer to the videos at https://warpyournoise.github.io/#warp-ldm.
>
> Other examples in the paper also highlight that transformations that are applied to the noise will produce similar image transformations, even though that’s not explicitly enforced during training of diffusion models. With this in mind, we believe that our proposed temporally coherent noise would indeed improve the training of video diffusion models, similar to PYoCo [1].
>
> ---
>
> **Sensitivity with respect to flow map**
>
> > I’m wondering how sensitive the method is with respect to the underlying estimated flow map? In other words, how do the errors in the estimated optical flows affect the results?
>
> Overall, as one would expect, accurate optical flow is important for achieving good results. From our experiments, we observed that poorly estimated optical flow will still be able to warp the noise, but it tends to create visual artifacts in the form of temporal misalignment, i.e. fine details in the video will not seem to be perfectly synchronized with the motion of larger structures.
>
> That being said, it is worth noting that the method is still relatively robust to small errors. For the video super-resolution and JPEG restoration task we show in the submission (Figure 4), the optical flow is computed on the degraded, low-resolution video, and our method is still able to produce decent, coherent results.
>
> Lastly, as our noise warping method is agnostic to the way the deformation field is computed, it can also work with more robust methods such as Neural Layered Atlases [2], when flow-based methods struggle to provide accurate motion estimation. These methods map each pixel in each frame to a canonical space common to all frames. In this case, the canonical space is considered the “first frame” and the mapping replaces the accumulated flow map in our algorithm. Please refer to our answer to Reviewer 3 for a more complete discussion about the properties of the warping velocity field.
>
> ---
>
> **Comparison with DDIM inversion**
>
> > I’m wondering how will the noise maps obtained [using DDIM inversion] compare to the ones obtained from the $\int$-noise in terms of the final video quality?
>
> The main issue that prevents a fair and straightforward comparison with DDIM inversion is that the latter only produces one single noise map per image, whereas the applications we show generally use DDPM and require as many noise maps as there are steps in the denoising process.
>
> We included a new visual comparison for the bedroom SDEdit surrogate rendering results with DDIM noise inversion that can be visualized here: https://warpyournoise.github.io/#ddim. There are two ways we can think of to use DDIM inversion with SDEdit.
> * The first consists in partially inverting the image to an intermediate noisy version using DDIM inversion and denoising it with forward DDIM. This expectedly leads to a reconstruction of the input synthetic scene with no additional realistic details.
> * The second way consists in fully inverting each frame to obtain a set of noise maps, which we treat similarly to the other noise priors we compare to in the original submission. Since the synthetic renders we use as input video are far from the distribution of the LSUN dataset, the inverted noises are not Gaussian. Our results demonstrate this creates noticeable artifacts.
>
> Lastly, DDIM inversion deterministically maps each frame to a noise, which removes any diversity in the results. In comparison, our method can apply the same warping to different noise maps to generate different equally temporally-consistent appearances.
>
> ---
>
> **Conclusion / References**
>
> We hope this clarified some of the interrogations. We will update the paper to integrate the feedback. Please feel free to reach out for any further questions/comments regarding the submission and our answers above.
>
> References:
>
> [1] Ge, Songwei, et al. "Preserve your own correlation: A noise prior for video diffusion models." Proceedings of the IEEE/CVF International Conference on Computer Vision. 2023.
>
> [2] Kasten, Yoni, et al. "Layered neural atlases for consistent video editing." ACM Transactions on Graphics (TOG) 40.6 (2021): 1-12.

---

> > ### Comment · Reviewer_13WQ · 2023-11-18
> >
> > I thank the authors for addressing my comments. I have no further questions at this point.

---

### Author Response · Authors · 2023-11-22
**Thank you for your feedback**

We would like to thank the reviewers for their feedback and questions. In the past few days, we have replied individually to all the comments and did our best to address the questions and suggestions. At the same time, we have set up a project webpage and added many new video results, including DDIM inversion results and some early integration with DeepFloyd IF. The webpage can be found here: https://warpyournoise.github.io. Based on these exchanges, we have also updated our manuscript with more details. Specifically, we have modified the following parts:

* Fig. 5.: replaced with a better visualization of the fluid super-resolution results
* Table 1: added evaluation of sample quality for the appearance transfer example with Improved Precision [1]
* Appendix C.3: added the derivation from continuous formulation to discrete
* Appendix C.4: clarified some technical details about our algorithm, in particular about the treatment of disocclusions.
* Appendix C.6: added a section comparing noise warping methods
* Appendix C.7: added runtime comparisons with baselines
* Appendix D.1: additional results comparing with DDIM inversion (see also https://warpyournoise.github.io/#ddim)

We hope the reviewers will find these changes helpful, and we remain open to more feedback and suggestions.


References:

[1] Kynkäänniemi et al. Improved Precision and Recall Metric for Assessing Generative Models. In NeurIPS, 2019.

---

### Meta-Review · Area_Chair_AWQ1 · 2023-12-13

**Metareview:**

This paper introduced a problem "how to properly warp the Gaussian noise such that the warped noise map still has the Gaussian distribution preserved?" and came up with a mathematically grounded solution. The practical value of this problem is temporally consistent video editing by image diffusion models.
Strengths: (1) the introduction of the problem, (2) the solution, (3) the verification of the practical value for video editing.
Weaknesses: (1) the assumption that temporally correlated noise maps can induce temporally consistent video editing results which doesn't have proof and is still an empirical observation, (2) the method is computationally inefficient.
The camera-ready version should include the discussion on the weaknesses and other issues raised by the reviewers.

**Justification For Why Not Higher Score:**

N/A

**Justification For Why Not Lower Score:**

All reviewers gave very positive feedbacks. The paper introduces a new problem "how to properly warp the Gaussian noise such that the warped noise map still has the Gaussian distribution preserved?", gave a solution which is practically useful for temporally consistent video editing. The paper may have broader audience, e.g., video generation or 3d generation, or inspire the noise map design.

---

### Decision · Program_Chairs · 2024-01-16

Accept (oral)